# The value of supportive care: A systematic review of cost-effectiveness of non-pharmacological interventions for dementia

Angelica Guzzon[1,2], Vincenzo Rebba[1,3]*, Omar Paccagnella[4], Michela Rigon[5], Giovanni Boniolo[6]

1 CRIEP (Interuniversity Research Centre on Public Economics), Veneto, Italy, 2 Department of Economics, Ca' Foscari University of Venice, Venice, Italy, 3 Department of Economics and Management "Marco Fanno", University of Padova, Padova, Italy, 4 Department of Statistical Sciences, University of Padova, Padova, Italy, 5 OIC Foundation, Padova, Italy, 6 Department of Neuroscience and Rehabilitation, University of Ferrara, Ferrara, Italy

* vincenzo.rebba@unipd.it

## Abstract

### Background

Almost 44 million people are currently living with dementia worldwide. This number is set to increase threefold by 2050, posing a serious threat to the sustainability of healthcare systems. Overuse of antipsychotic drugs for the management of the symptoms of dementia carries negative consequences for patients while also increasing the health expenditures for society. Supportive care (SC) interventions could be considered a safer and potentially cost-saving option. In this paper we provide a systematic review of the existing evidence regarding the cost-effectiveness and cost-utility of SC interventions targeted towards persons living with dementia and their caregivers.

### Methods

A systematic literature review was performed between February 2019 and December 2021 through searches of the databases PubMed (MEDLINE), Cochrane Library, CENTRAL, Embase and PsycINFO. The search strategy was based on PRISMA 2020 recommendations. We considered studies published through December 2021 with no lower date limit. We distinguished between five categories of SC strategies: cognitive therapies, physical activity, indirect strategies (organisational and environmental changes), interventions primarily targeted towards family caregivers, and multicomponent interventions.

### Results

Of the 5,479 articles retrieved, 39 met the inclusion criteria. These studies analysed 35 SC programmes located at different stages of the dementia care pathway. Eleven studies provided evidence of high cost-effectiveness for seven interventions: two multicomponent interventions; two indirect interventions; two interventions aimed at caregivers of community-

**Data Availability Statement:** All relevant data are within the manuscript and its Supporting information files.

**Funding:** This study has been carried out within the research project "Supportive care for elderly people with severe cognitive impairment" sponsored by the OIC (Opera Immacolata Concezione) Foundation, Padova, Italy.

**Competing interests:** The authors have declared that no competing interests exist.

dwelling persons with dementia; one community-based cognitive stimulation and occupational programme.

## Conclusion

We find that the most promising SC strategies in terms of cost-effectiveness are multicomponent interventions (targeted towards both nursing home residents and day-care service users), indirect strategies (group living and dementia care management at home), some forms of tailored occupational therapy, together with some psychosocial interventions for caregivers of community-dwelling persons with dementia. Our results suggest that the adoption of effective SC interventions may increase the economic sustainability of dementia care.

## Introduction

Globally, about 43.8 million people were living with dementia in 2016 [1], and this number is projected to triple by 2050 [2]. The worldwide costs of Alzheimer's disease, the most prevalent subtype of dementia, and other dementias were estimated at US$818 billion in 2015 (equivalent to 1.09% of the global gross domestic product); these costs overcame the threshold of US $1 trillion in 2018 and are projected to double by 2030 [1, 3]. The substantial increase in the prevalence of dementia, mostly driven by demographic trends, poses significant challenges to health and social care systems, especially in terms of economic sustainability [3–6]. The scale of the problem becomes even greater when we take into account the indirect costs related to caregivers of persons with dementia (PwDs). Caregiving exacts a toll on caregivers' mental, emotional, physical, and financial health [7]; caregivers are twice as likely to suffer from depression [8], they use more medication and make more doctor visits [9, 10], they save less money, and up to 9% of caregivers need to quit their jobs [11].

Current drug therapies for dementia (cholinesterase inhibitors and memantine) has been shown to have a small effect on cognition. However, these medications do not significantly change the course of illness and may provoke side effects [2, 12, 13], while only a limited number of ongoing clinical trials are currently investigating the viability of drugs directed at diverse therapeutic targets [13–15]. Available medications for treating PwDs have been shown to enhance the quality of life (QoL) for both the patient and caregiver when prescribed at the appropriate time during illness. In particular, cholinesterase inhibitors are more cost-effective than placebo and probably also cost-saving (by delaying the onset of institutionalisation), while the evidence in support of the cost-effectiveness of combination therapy (a cholinesterase inhibitor plus memantine) is less clear [16, 17]. A new drug, aducanumab, has been recently approved by the US Food and Drug Administration for treatment of Alzheimer disease, even though it is uncertain whether it works at all or provides sufficient benefits to outweigh its harms [18]. While there are uncertainties relating to the cost-effectiveness of some of the drugs for dementia, there are potentially serious risks associated with using antipsychotic medications to treat the behavioural and psychological symptoms of dementia (BPSD) [19–21]; moreover, there is no clinical or economic case for using antidepressant drugs to treat people with Alzheimer's disease who have comorbid depression [17]. A problem with antipsychotic medications is that these drugs do not offer a sufficient benefit relative to the risks they pose [22]. For example, a UK study shows that patients who received an antipsychotic treatment for 12 months were significantly more likely to have died by the 24-month and

36-month follow-up periods compared to patients who had received a placebo [23], while other studies have found a link between the use of antipsychotic drugs in dementia patients and an increase in the risk of acute pulmonary diseases, hip fracture, thromboembolism, and stroke [24–27]. Despite this evidence in conjunction with the warnings and best practice guidelines that have followed [19–21], the use of antipsychotic drugs is still widespread.

Considering the setbacks suffered in the research on viable pharmacological treatments to counteract the progression of different types of dementia [12–15], the serious risks associated with using antipsychotic medications [19–21], and the high costs of overprescribing anti-dementia drugs [28], the development of effective non-pharmacological interventions to integrate or substitute the use of medications is of particular importance to increase both the effectiveness and the economic sustainability of dementia care.

Given this background, resource allocation could be enhanced by shifting from a standard approach—focused on containing the impact of distressing symptoms on patients through medications—to a more comprehensive approach based on the notion of person-centred care. This new approach would also follow the patient throughout the whole course of the disease by providing personalised care as well as support to patients and families. Defined as *supportive care* [29] and representing "a full mixture of biomedical dementia care, with good quality, person-centred, psychosocial, and spiritual care," this approach must be extended throughout the course of the illness to guarantee the overall wellbeing of PwDs and their caregivers [30]. The term supportive care (SC) refers to a wide array of non-pharmacological interventions that encompass a broad and growing range of services that are delivered either to the patient, the caregiver, or the patient-caregiver dyad [31, 32]. This approach has been previously experimented in cancer care for addressing the clinical and psychosocial needs of patients in order to provide optimal quality of life [33] and in end-of-life care for non-cancer patients [34]. One of its key aspects is the decreasing reliance on medications that do not offer a sufficient benefit relative to the risks they pose in favour of novel non-pharmaceutical interventions [35]. In the case of PwDs, SC is characterised by the continuous assistance of patients and their relatives from diagnosis until death, a holistic and interdisciplinary approach to care, and a high level of flexibility in choosing the right care practices for each case [36]. It is therefore evident that this definition of SC should not be confused with the one sometimes adopted in the cost-effectiveness literature, where the term "best supportive care" is used to denote care as usual or non-intervention.

A key feature of SC in all its stages is the central role of both formal and informal caregivers: the former are meant to have in-depth knowledge and competencies to deal with dementia patients, while the latter need to be recognised as indispensable players in dementia care, and both need to form and maintain collaborative relationships to guarantee high-quality care to patients [37]. In particular, support to caregivers could be considered a win-win solution, as it is beneficial for carers, patients, and the sustainability of healthcare systems [22]. On the opposite side of the spectrum, the unregulated use—and sometimes abuse—of antipsychotic drugs is a no-win situation, as it is detrimental for the health of the patient, and it puts a strain on the budget of healthcare systems [35]. In particular, regarding the management of behavioural problems in Alzheimer's disease, Gauthier et al. [38] suggest that non-pharmacological interventions (including psychosocial/psychological counselling as well as interpersonal and environmental management) should be attempted first, followed by the least harmful medication for the shortest time possible [38].

A unified classification of SC interventions for dementia does not exist yet, but several taxonomies can be found in the literature. For instance, Cammisuli et al. distinguish between holistic techniques, brief psychotherapy, cognitive methods, and alternative strategies [39]; in contrast, D'Onofrio et al. distinguish between cognitive and emotion-oriented interventions,

sensory and multi-sensory stimulation interventions, and other interventions [40]. The World Alzheimer Report 2011 [41] and Nickel et al. [42] classified non-pharmacological interventions into four categories: physical exercise, interventions to support and enhance cognitive abilities, psychological and behavioural therapies, and occupational therapy. Our analysis will consider five categories of SC interventions for dementia, as we will explain in next section.

Over the last few years, the evidence base on the effectiveness of non-pharmacological interventions (in terms of cognitive functioning and the reduction of behavioural symptoms of PwDs) has grown considerably [31, 32]. Conversely, evidence on the value for money of non-pharmacological and SC interventions for PwDs and their caregivers is still scant, despite the growing need for healthcare systems to base resource allocation decisions on cost-effective intervention strategies.

In this paper, we provide a systematic review of the main evidence on the cost-effectiveness of five different categories of non-pharmacological and supportive practices for dementia. Our analysis aims at identifying which types of SC intervention have shown the strongest evidence of cost-effectiveness in order to provide useful information for the design of policies which may increase the economic sustainability of dementia care. Previous systematic reviews have highlighted the scarcity of economic evidence on non-pharmacological interventions for PwDs [43, 44] and their caregivers [45]. These reviews were focused mainly on interventions for community-dwelling persons with mild to moderate dementia [42] and home support interventions [46]. In contrast, we try to offer an all-encompassing review of cost-effectiveness studies on non-pharmacological and psychosocial interventions that target PwDs, their caregivers (either formal or informal), or the patient-caregiver dyad, and which are located at different stages of the care pathway for dementia and in different settings. Moreover, our systematic review is more complete since it provides evidence on the cost-effectiveness of indirect interventions such as organisational changes and innovations in the delivery of care and support.

## Materials and methods

### Categories of supportive care interventions

In this paper, we distinguish between five categories of non-pharmacological/SC strategies: 1) cognitive therapies; 2) physical activity interventions; 3) indirect strategies; 4) interventions primarily targeted towards caregivers; 5) and multicomponent interventions.

The main characteristics of the interventions considered are described in Table 1.

### Search strategy and criteria for inclusion

A systematic literature review was performed between February 2019 and December 2021 on the healthcare electronic databases MEDLINE (PubMed), CDSR (Cochrane Database of Systematic Reviews), CENTRAL (Cochrane Central Register of Controlled Trials), Embase and PsycINFO. The search terms used to identify the articles to include in the review were as follows: (dementia OR alzheimer* OR cognitive) AND ('cost-effectiveness'/exp OR 'cost-analysis'/exp OR 'cost-utility'/exp) AND ('non-pharmacological'/exp OR psychosocial* OR 'drug-free'/exp). We considered studies published through December 2021 with no lower date limit. Additional details on the electronic search strategy can be found in the S1 File.

Study eligibility was based on the following criteria:

- Studies evaluating non-pharmacological dementia interventions;

- Interventions aimed at either the patient or the caregiver (or the dyad patient-caregiver);

**Table 1. Categories of supportive care interventions.**

| Category | General description | Examples of SC intervention | Specific description |
|---|---|---|---|
| 1) Cognitive therapies | All those methods that stimulate a patient's cognition and may also control BPSD in several ways, including cognitive stimulation and occupational therapy. | Cognitive Stimulation Therapy (CST) | An evidence-based rehabilitation technique to enhance residual cognitive abilities and functional skills and preserve implicit memory [47]; patients are involved in activities such as word association games, quizzes, number games, physical games, and creative activities [48]. |
| | | Occupational therapy | It has the primary focus of preserving patients' independence by improving their ability to perform ADLs and adapt to their living environment; it can also be administered in a home setting by trained caregivers [49]. |
| | | Reality Orientation Therapy (ROT) | One of the most popular psychosocial interventions to manage dementia, it has the main goal of spatially and temporally reorienting patients, but it also helps the patient to maintain social interaction [50, 51]. |
| | | Reminiscence therapy | It encourages patients to recall and talk about past experiences and events in their lives, either in individual or group sessions, and with the aid of props like photographs and videos [52]. |
| | | Learning therapy | A combined form of cognitive training and stimulation (adopted especially in Japan and the U.S.), where instructors help patients to perform simple calculations or reading tasks with face-to face verbal communication [53]. |
| | | Art or music therapy | Therapeutic use of art or music to provide a dementia patient with meaningful stimulation and improve her/his participation and level of self-esteem [54, 55]. |
| | | Intergenerational activities | Interaction between children and people with dementia to improve the patients' social interaction and sense of purpose [56]. |
| | | Doll and plushie therapy | Usually used on patients with advanced dementia, it engages the patient in behaviours such as holding, cuddling, feeding and dressing dolls or plushies [57]. |
| | | Pet therapy | Interaction of patients with animals, including activities such as petting, feeding, and playing with dogs and other animals [58]. |
| 2) Physical activity interventions | Interventions that can produce health benefits for patients, such as decreasing the number of falls and improving sleep and mood. | Aerobic exercise | Walking, cycling and gymnastics [59]. |
| | | Mixed exercise | Aerobic exercise and Resistance training [59]. |
| | | Dance therapy | Dance sessions with PwD combine exercise with creative expression and recreation activities [60]. |
| 3) Indirect strategies | Strategies that include organisational and environmental changes, together with innovations in the delivery of care and support. | Dementia care management programmes | Interventions delivered in the community aiming to coordinate the treatment and care for PwDs with respect to their needs and the recommendations of evidence-based guidelines [61–63]. |
| | | Dementia Care Mapping (DCM) | Observational tool that assists in the delivery of better formal care to PwDs, allowing for the adoption of a person-centred care approach to improve the quality of care of dementia patients [64]. |
| | | Managing Agitation and Raising Quality of Life (MARQUE) | Manual-based intervention targeted at the staff of care homes, is designed to train them in the implementation of procedures to reduce agitation in dementia patients [65]. |
| | | Memory clinics | Facilities that provide guidance, prescriptions, rehabilitation, and various non-pharmacological interventions to dementia patients [66]. |
| | | Group living | Interventions that allow patients with a similar level of cognitive impairment to cohabite in a controlled environment [67]. |
| | | Assistive technology, telehealth and telecare | Electronic or mechanical devices that can support independence and improve quality of life by assisting with daily living activities, reducing harmful risks and improving communication [68]. |
| 4) Interventions primarily targeted at family caregivers | Interventions with the primary goal to reduce the burden of care on the family caregivers of a PwD. | Respite care | Any kind of arrangement that provides short-term relief to primary caregivers by providing the patient with an alternative source of care and supervision; adult day centres and nurse visits are both forms of respite care [69, 70]. |
| | | Programmes for caregivers | Any intervention, usually of a psychosocial nature, that is primarily aimed at the primary caregivers; examples are support groups, family meetings, and coping strategies [71–75]. |
| | | Assistive technology, telehealth and telecare | Telehealth support that links family caregivers to tailored feedback from dementia care experts based on caregiver-initiated video recordings of challenging care situations [76]. |

*(Continued)*

**Table 1.** (Continued)

| Category | General description | Examples of SC intervention | Specific description |
|---|---|---|---|
| 5) Multicomponent interventions | Protocols that combine two or more different interventions included in categories 1, 2, 3 and 4. | Wellbeing and Health for People with Dementia (WHELD) | Implemented within a person-centred care framework, it includes physical exercise, approaches to reduce agitation, and psychosocial activities [77, 78]. |
| | | Integrated approaches | Personalised bundles of non-pharmacological interventions for the patient-caregiver dyad that are chosen after mapping their needs [79, 80]. |
| | | Multicomponent support programmes | Targeted at couples for whom one of the spouses suffers from dementia, they are support programmes that include group meetings for the caregivers, scheduled assessments by a geriatrician, and individualised services for the couple [81]. |
| | | Journeying through Dementia (JtD) | Targeting the early stages of dementia, it combines occupational therapy with self-management and peer support [82]. |
| | | Motor, ADL, Cognitive and Social functioning stimulation (MAKS) | Group-based intervention that targets patients in day care centers and which includes cognitive stimulation, ADL activation, sensory and social stimulation [83]. |
| | | Namaste Care Family program | Person-centered care intervention for advanced dementia which combines psychosocial, sensory and spiritual components [84]. |

- The participants in the study had a diagnosis of dementia or were caregivers of a person with diagnosed dementia;

- Research conducted as randomised controlled trial (RCT) or prospective cohort study;

- Studies with a complete economic analysis including an economic evaluation (a cost-effectiveness analysis and/or cost-utility analysis) or at least comprehensive information on outcomes and costs [85];

- Studies with an abstract in English.

   We also considered the following exclusion criteria:

- Studies on the ageing population with no explicit focus on dementia;

- Research conducted as retrospective study;

- Studies which did not show a complete economic analysis upon full text screening;

- Studies with no available abstract in English.

   The search strategy and the following review are both based on PRISMA 2020 recommendations [86]. The outcomes of the study selection process are described in the Results section. No prespecified protocol was followed for this systematic review.

## Data collection and analysis

Data extraction was performed according to the guidelines of the Centre for Reviews and Dissemination for reviews of economic evaluations [87]. After the removal of duplicate citations using Endnote X9, the titles and abstracts of the remaining articles were initially screened by the principal reviewer (AG). After the initial screening, the principal reviewer (AG) evaluated the abstracts of the remaining publications and applied the eligibility criteria with another member of the research team (VR). Two members of the research team (AG and VR) performed the full-text screening on the publications that met the eligibility criteria. Information was collected on the type of economic evaluation, study objective, study design, description of the intervention, comparators, measures of benefit and cost, and outcome and cost results. Summary results were independently presented to the other members of the research team (OP, MR, and GB) to solve any disagreement through discussion or consultation.

The high heterogeneity in terms of interventions and outcome measures of the studies evaluated made it impossible to perform a meta-analysis, so we proceeded with a qualitative analysis.

## Quality appraisal of included studies

The methodological quality of the studies included in the review was assessed using the Consensus on Health Economic Criteria (CHEC) checklist [88], which consists of 19 yes-or-no questions. To each study, we assigned a score from 0 to 19 based on the number of questions that the assessor answered with a "yes". Studies were classified as being high-quality if the score was equal to or higher than 17, medium-quality studies were those with a score between 14 and 16, and low-quality studies were those which scored 13 or lower. The score also reflects the information contained in additional analyses for those that actively pointed to other articles for additional information on the study design and/or protocol. The principal reviewer (AG) assessed the quality of all the articles, and the other four members of the research team (VR, OP, MR, and GB) checked for accuracy within their subsets. Any disagreement was resolved through discussion or consultation. The quality appraisal was undertaken to aid in interpreting the findings and determining the strength of the conclusions drawn; no study was excluded based on the results of the quality assessment.

## Main outcomes of supportive care interventions

As mentioned above, the outcome indicators considered in the studies included in the review were highly heterogeneous even though it is possible to broadly distinguish between patient and caregiver outcome measures.

The main patient outcomes considered in the analysed studies were the following:

- Quality of life–the cognitive and functional decline brought about by dementia has a huge impact on the patient's QoL, and most studies include both generic health-related QoL (HRQoL) and dementia-related QoL as outcomes [89];

- Cognitive impairment–dementia impacts short- and long-term memory but also other cognitive functions such as language, abstract thinking, and judgement [90];

- Dementia severity–the gradual progression of the disease is measured with staging instruments that monitor the clinical and cognitive deterioration caused by dementia [91];

- Behavioural and psychological symptoms of dementia (BPSD)–these neuropsychiatric disturbances, such as apathy or hallucinations (or other non-cognitive symptoms), constitute a major component of dementia and have an impact on the QoL [92];

- General health–this variable is gauged by looking at comorbidities, adverse events (i.e., untoward medical occurrences in a patient, including falls and fractures), nutritional status, etc. [93, 94];

- Mental health–this variable can be measured by looking at an individual's depression levels, anxiety levels, schizophrenic or psychotic episodes, etc. [95];

- Agitation–as one of the most commonly observed neuropsychiatric symptoms in patients suffering from dementia, this condition is described as restless behaviour or improper physical and/or verbal action that can be a source of trouble for others [96];

- ADLs and IADLs–the number of (instrumental) activities of daily living an individual is able to carry out in an accepted way is a measure of functional capacity, which is an important indicator of health in the elderly [97];

- Prescription drug use–the use of antipsychotic medications to treat the BPSD;

- Service utilisation (and related costs)–the extent to which dementia patients use medical and/or social services and resources [98], including institutionalisation [99].

For each patient outcome, Table 2 reports the correlate measures considered in the selected studies.

The main caregiver outcomes considered in the reviewed studies were the following (see Table 3 for details on the different measures for each outcome):

- Quality of life–dementia severely impacts the QoL of caregivers because caring for someone who suffers from dementia is extremely burdensome and contributes to physical and psychiatric illnesses [100];

- Burnout and burden–caregiver burden is the perceived negative effect of caring for a family member [101], while caregiver burnout is more specifically a state of physical, emotional, and mental exhaustion [102];

- Sense of competence and mastery–competence is the extent to which a caregiver feels he or she can effectively do what is needed for a patient, whereas mastery is the extent to which a caregiver feels in control of the situation; both have been linked to positive outcomes for the caregiver [103, 104];

- General health–caregivers are more likely to report poor health because they have less time to take care of themselves and face substantial stress (as indicated by the increased levels of cortisol) [105, 106];

- Mental health–depression is very common among dementia caregivers, as are sleep disturbances, loneliness, and social isolation [107, 108];

- Quality of interaction with the patient–low-quality interactions can undermine both the caregiver's QoL and quality of care [109]; the quality of the relationship that occurs between the caregiver and the patient has been found to be predictive of outcomes like the patient's institutionalisation and functional decline [110, 111];

- Coping strategies–coping strategies employed by caregivers, such as avoidance or wishful thinking, are linked to physical and mental health outcomes [112];

- Time spent caregiving–caring for a PwD is not only a burdensome task, but it is also time consuming, as it prevents informal caregivers from having a regular work-life balance [113];

- Service utilisation–the additional medical and social service use by caregivers themselves helps us better understand the impact dementia has on societal costs [114];

- Absenteeism–formal and informal dementia caregivers are more likely to have higher absenteeism rates [115].

## Results

### Study selection

The systematic search identified 5,479 publications. Duplicate citations were removed using Endnote X9, resulting in a total of 1,362 publications. After an initial screening of the titles and abstracts, 229 publications remained. After applying the eligibility criteria, 55 publications remained for full-text screening. Of these, 16 articles were excluded due to specific issues (e.g., 7 articles were excluded since they did not report a complete economic analysis upon a full-

**Table 2. Patient outcomes and their measures in the reviewed studies.**

| Outcome | Measures |
|---|---|
| Health Related Quality of life (HRQoL) | • EuroQol (EQ-5D)<br>• Short Form-12 Health Survey (SF-12)<br>• Index of Well-Being (IWB)<br>• Rosser index |
| Dementia-Related Quality of life | • Dementia Quality of Life (DEMQOL)<br>• Quality of Life in Alzheimer's Disease (QoL-AD) |
| Cognitive impairment | • Mini-Mental State Examination (MMSE)<br>• Alzheimer's Disease Assessment Scale–Cognitive Subscale (ADAS-COG)<br>• Verbal fluency test (VF)<br>• Clock drawing test (CDT)<br>• Frontal Assessment Battery (FAB)<br>• Autobiographical Memory Interview (AMI) |
| Dementia severity | • Clinical Dementia Rating (CDR)<br>• Functional Assessment Staging of Alzheimer's Disease (FAST) |
| Behavioural and psychological symptoms | • Neuropsychiatric Inventory (NPI)<br>• Behavioural and Psychological Symptoms of Dementia (BPSD)<br>• Revised Memory and Behavior Problems Checklist (RMBPC) |
| General health | • Short Form-12 Health Survey (SF-12)<br>• General Health Questionnaire (GHQ)<br>• Charlson Comorbidity Index (CCI)<br>• Disability Assessment for Dementia (DAD)<br>• Falls and fractures<br>• Institutionalisation rates<br>• Certification of Needed Long-Term Care (CNLTC)<br>• COOP WONCA Functional Status Assessment Charts<br>• Mini Nutritional Assessment (MNA) |
| Mental health | • Hospital Anxiety and Depression Scale (HADS)<br>• Global Deterioration Scale (GDS)<br>• Cornell Scale for Depression in Dementia (CSDD)<br>• Rating of Anxiety In Dementia (RAID)<br>• MOS 20-Item Short Form Survey Instrument–Mental Health (MOS-20MH) |
| Agitation | • Cohen-Mansfield Agitation Inventory (CMAI) |
| Activities of daily living | • Alzheimer's Disease Cooperative Study ADL Scale (ADCS-ADL)<br>• Barthel Index (BI)<br>• Bristol Activities of Daily Living Scale (BADLS)<br>• Groningen Activities Restriction Scale (GARS)<br>• Lawton Brody scale (IADLs)<br>• Assessment of Motor and Process Skills (AMPS)<br>• Interview for Deterioration in Daily Living Activities in Dementia (IDDD)<br>• Katz scale (ADLs)<br>• Multi-Dimensional Dementia Assessment Scale (MDDAS) |
| Use of prescription drugs | • Use of antipsychotics |
| Service utilisation | • Client Service Receipt Inventory (CSRI)<br>• Resource Utilization in Dementia (RUD)<br>• Institutionalization rates |

**Table 3. Caregiver outcomes and their measures in the reviewed studies.**

| Outcome | Measures |
|---|---|
| Health Related Quality of Life (HRQoL) | • EuroQol (EQ-5D)<br>• Short Form-12 Health Survey (SF-12)<br>• World Health Organization Quality of Life Brief Version (WHOQoL-BREF)<br>• RAND 36-Item Health Survey (RAND-36) |
| Care Related Quality of Life | • Caregiver Quality of Life Instrument (CQLI) |
| Caregiver burden and burnout | • Maslach Burnout Inventory (MBI)<br>• Zarit Burden Interview (ZBI) |
| Sense of competence and mastery | • Sense of Competence in Dementia Care (SCID)<br>• Sense of Competence Questionnaire (SCQ)<br>• Pearlin Mastery Scale (PMS) |
| General health | • Short Form-12 Health Survey (SF-12) |
| Mental health | • Hospital Anxiety and Depression Scale (HADS)<br>• Relative Stress Scale (RSS)<br>• Mini International Neuropsychiatric Interview (MINI)<br>• Centre for Epidemiologic Studies Depression Scale (CES-D)<br>• State Trait Anxiety Inventory (STAI) |
| Quality of relationship | • Quality of Interactions Schedule (QUIS)<br>• Quality of Carer and Patient Relationship scale (QCPR) |
| Coping strategies | • COPE inventory |
| Time spent caregiving | • Caregiving time spent doing things<br>• Caregiving time spent being on duty<br>• Resource Utilisation in Dementia questionnaire |
| Service utilisation | • Health Services Utilization Questionnaire (HSUQ) |
| Absenteeism | • Time away from work |

text screening). A final sample of 39 studies remained for inclusion in the review, including a study in the Dutch language [79]. The search strategy, based on PRISMA recommendations [86], is shown in the flow chart in Fig 1.

## Characteristics of the included studies

We reviewed 39 studies that analysed 35 interventions: nine cognitive stimulation and occupational programmes primarily targeted at PwDs; three physical activity interventions; ten indirect interventions (organisational and environmental changes); seven interventions primarily targeted towards family carers; and six structured multicomponent interventions. The number of reviewed study (39) is higher than the number of analysed interventions (35) since some studies considered the same intervention but with a different timeframe or a different set of outcome measures: two studies [63, 116] focused the same intervention of dementia care management (Delphi-MV trial); three studies [71, 117, 118] considered the same individual therapy program for caregivers "Strategies for Relatives" (START); two studies [77, 78] considered the multicomponent WHELD ("Improving Wellbeing and Health for People with Dementia") intervention. The main characteristics of the studies are summarised in Tables 4–8.

The interventions analysed in the studies were located at different stages of the care pathway for dementia: 14 studies focused on patients with dementia in its mild to moderate stages

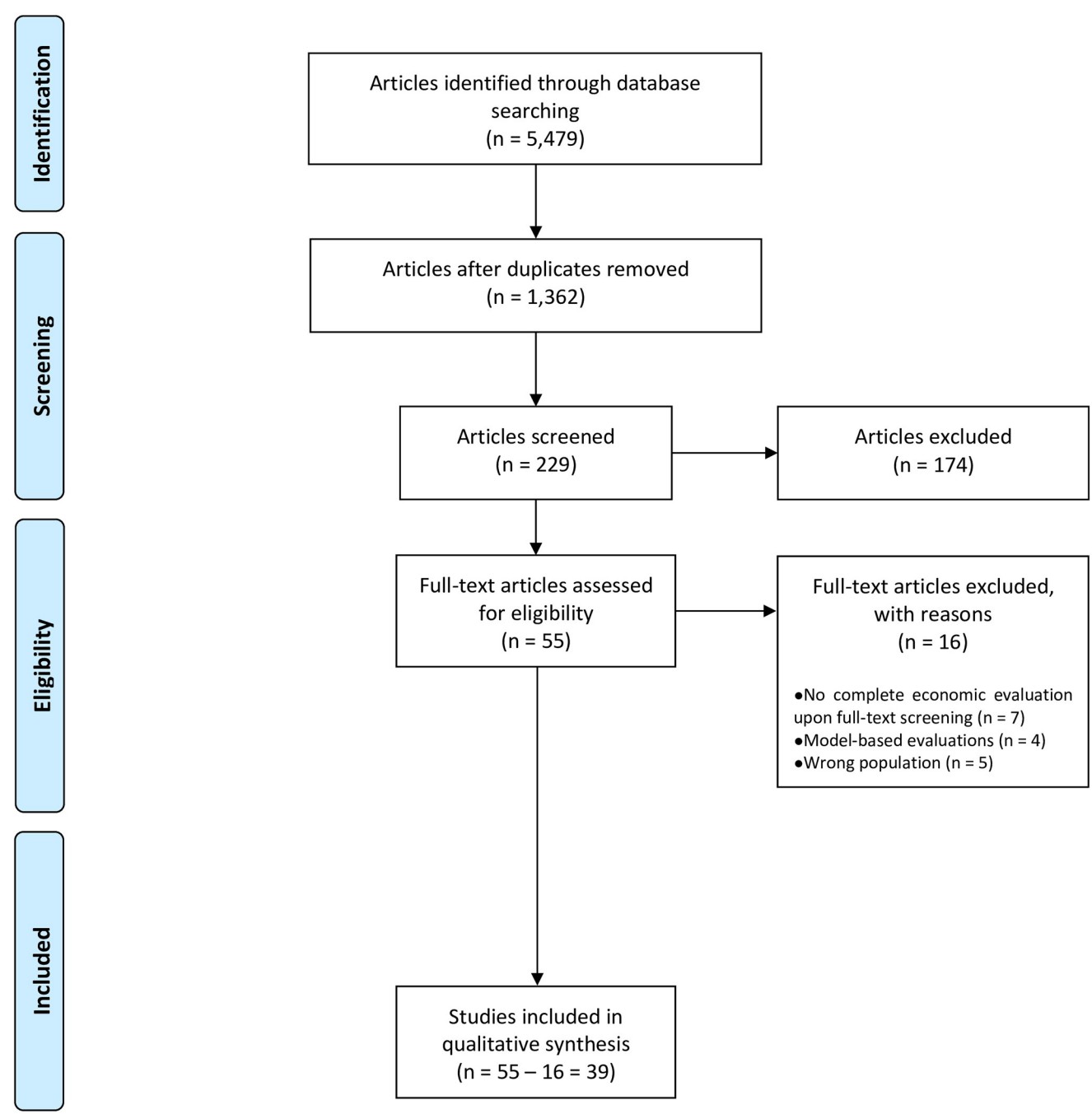

**Fig 1. PRISMA flow chart of the study selection process.**

and/or their caregivers [63, 66, 72, 80, 83, 116, 119–126]; 19 studies addressed the moderate-to-severe stages [65, 67, 69–71, 73, 77–79, 84, 117, 118, 127–133]; and six studies focused on PwDs at different stages [62, 68, 76, 81, 134, 135]. Eight studies considered SC programmes in nursing homes and assisted living settings [65, 77, 78, 84, 128–130, 133], and two studies analysed both residential and community settings [122, 134], while the rest of the health economic analyses concerned community-based interventions. Most studies analysed SC interventions

**Table 4. Main characteristics of studies evaluating the cost-effectiveness of cognitive therapy interventions.**

| No. | Study | Intervention description and comparator (Setting) | Country | Type of study and economic evaluation (Time horizon) | Sample size No. PwDs/No. Caregivers (Intervention Group/Control Group) | Perspective | PwD outcome measures | Caregiver outcome measures | Incremental Cost-Effectiveness Ratio (ICER) and other cost-effectiveness measures | Cost-effectiveness assessment | Cost-effectiveness rationale | Quality of the study |
|---|---|---|---|---|---|---|---|---|---|---|---|---|
| 1 | Graff et al. 2008 [119] | Community occupational therapy (including cognitive and behavioural interventions) vs Usual Care (Community-based: Memory clinics, Day clinics of a geriatrics department, home) | Netherlands | RCT CEA (6 weeks; 3 months) | 135 / 135 (68 / 67) | Societal | • Daily functioning (AMPS; IDDD) | • Sense of competence (SCQ) | €1,748 saved compared with control (difference in mean total care costs per successful treatment) | High | Dominant intervention (better outcomes and lower costs than comparator) with a probability of 95%. | High |
| 2 | Clare et al. 2019 [121] | Cognitive rehabilitation (GREAT trial) vs Usual Care (Community-based; home) | UK | RCT CEA+CUA (3 months; 6 months) | 427 / 427 (209 / 218) | Health and social care system Societal | • Self-reported goal attainment (BGSI) • Quality of life (DEMQOL) • Depression and anxiety (HADS) • Self-efficacy (GSES) • Cognitive impairment measures • Service utilisation (CSRI) | • Relatives' Stress Scale (RSS) • Health status assessment • Quality of life (EQ-5D, WHOQoL-BREF) | Point estimates from the health and social care perspective: £1,296 for an increase of 1.32 points in the BGSI attainment rating; £1,110,000 per QALY gained (patient), £632,000 per QALY gained (caregiver) | Moderate | Partial cost-effectiveness. The probability of cost-effectiveness in terms of participant-rated goal attainment (BGSI) was over 99% at a WTP of £2500 under both the health and social care and the societal perspectives. No evidence of cost-effectiveness in terms of gains for relevant outcomes and QALYs (for both PwDs and carers). | High |
| 3 | D'Amico et al. 2015 [122] | Maintenance cognitive stimulation therapy vs Usual Care alone (Different Settings: Nursing Home and Community Centre) | UK | RCT CEA + CUA (24 weeks) | 199 / 0 (106 / 93) | Health and social care system Societal | • Cognitive impairment (ADAS-Cog, MMSE) • Quality of life (QoL-AD, DEMQOL, EQ-5D) • Behavioural and psychological symptoms (NPI) • ADLs (ADCS-ADL) • Service utilisation (CSRI) | — | £266 per QoL-AD point; £26,835 per proxy-rated QALY; £558 per MMSE point | Moderate | Potential cost-effectiveness in terms of self-rated QoL-AD (primary outcome) with a probability of 90% at a WTP of £1,400 (but the authors specify that there are no established WTP thresholds for QoL-AD). Low cost-effectiveness in terms of secondary outcomes: low probability that the ICER is within the range £20,000-£30,000 per QALY associated with NICE recommendations; partially cost-effective intervention in terms of MMSE score, but not in terms of ADAS-Cog score. | Medium |

*(Continued)*

**Table 4.** (Continued)

| No. | Study | Intervention description and comparator (Setting) | Country | Type of study and economic evaluation (Time horizon) | Sample size No. PwDs/No. Caregivers (Intervention Group/Control Group) | Perspective | PwD outcome measures | Caregiver outcome measures | Incremental Cost-Effectiveness Ratio (ICER) and other cost-effectiveness measures | Cost-effectiveness assessment | Cost-effectiveness rationale | Quality of the study |
|---|---|---|---|---|---|---|---|---|---|---|---|---|
| 4 | Orgeta et al. 2015 [123] | Carer-led individual cognitive stimulation therapy vs Usual Care (Community-based: Memory clinics and community mental health teams for older people) | UK | RCT CEA+CUA (13 weeks; 26 weeks) | 273 / 273 (134 / 139) | Health and social care system Societal | • Cognitive impairment (ADAS-Cog, MMSE) • Quality of life (QoL-AD, DEMQOL-Proxy) • Behavioural and psychological symptoms (NPI) • BADLS • Depression and anxiety (GDS), • Relationship (QCPR) • Service utilisation (CSRI) | • Mental and physical health (SF-12) • Depression and anxiety (HADS) • Distress (NPI) • Quality of life (EQ-5D) • Carer resilience (RS-14) • Relationship (QCPR) • Service utilisation (CSRI) | £3,100 per QALY gained (caregivers) | Moderate | Partial cost effectiveness: cost-effective intervention only in terms of caregiver's QoL. The probability of the intervention being cost-effective at a WTP per QALY corresponding to the NICE's threshold of £30,000 was 81% from the health and social care perspective and 93% from the societal perspective. | High |
| 5 | Gitlin et al. 2010 [127] | Tailored Activity Program (TAP) for patients and caregivers (occupational therapy) vs Usual Care (Wait-list) (Community-based: home) | USA | RCT CEA (4 months) | 60 / 60 (30 / 30) | Societal | — | • Caregiving time spent "doing things" • Caregiving time spent "being on duty" | $2.37/day to save 1 hour of caregiving time "doing things" for PwD; $1.10/day to save 1 hour of caregiving time "being on duty" | Moderate | Partial cost-effectiveness: intervention is cost-effective 79% of the time for the outcome measure "doing things" and 79.6% of the time for "on duty" based on an individuals' WTP threshold of $3,893 per person (potential financial savings obtained over a 4-month time). No evidence of cost-effectiveness for PwDs due to the absence of primary outcomes for patients. | Low |
| 6 | Sado et al. 2020 [128] | Learning Therapy vs Usual Care (Nursing home) | Japan | Prospective study CBA (12 months) | 57 / 0 (30 / 27) | Health and social care system | • Level of care needed (CT-CNLTC) • Quality of life (EQ-5D) • Cognitive impairment (MMSE, FAB) • PMS/IADL | — | A yearly net monetary benefit per patient of US $1,605 (90.8% probability of the net monetary benefit being positive) | Moderate | Partial cost-effectiveness: cost savings in terms of lower levels of care needed due to improved patients' function of daily living (measured with CT-CNLTC); no evidence of cost-effectiveness in terms of gains for relevant outcomes and QALYs. | Medium |

*(Continued)*

**Table 4.** (Continued)

| No. | Study | Intervention description and comparator (Setting) | Country | Type of study and economic evaluation (Time horizon) | Sample size No. PwDs/No. Caregivers (Intervention Group/Control Group) | Perspective | PwD outcome measures | Caregiver outcome measures | Incremental Cost-Effectiveness Ratio (ICER) and other cost-effectiveness measures | Cost-effectiveness assessment | Cost-effectiveness rationale | Quality of the study |
|---|---|---|---|---|---|---|---|---|---|---|---|---|
| 7 | Knapp et al. 2006 [134] | Cognitive stimulation therapy vs Usual Care (Different Settings: Nursing Home, Community and Day Centre) | UK | RCT CEA (8 weeks) | 161 / 0 (91 / 70) | Health and social care system | • Cognitive impairment (MMSE) <br> • Quality of life (QoL-AD) <br> • Service utilisation (CSRI) | — | £75.32 per MMSE point, £22.82 per QoL-AD point (point estimates) | Moderate | The intervention is potentially cost-effective considering both the cognition outcome and the QoL-AD measure. However, the authors underline that evidence of actual WTP for cognitive improvements and QoL-AD gains is very limited. | Medium |
| 8 | Woods et al. 2012 [124] | Reminiscence group therapy vs Usual Care (Community-based: Memory clinics and Community mental health teams for older people) | UK | RCT CEA (10 months) | 350 / 350 (206 / 144) | Health and social care system | • Quality of life (QoL-AD, EQ-5D) <br> • Autobiographical memory (AMI (E)) <br> • Quality of relationship (QCPR) <br> • Depression and anxiety (CSDD, RAID) <br> • BADLs <br> • Service utilisation (CSRI) | • Mental health (GHQ-28) <br> • Quality of life (EQ-5D) <br> • Quality of relationship (QCPR) <br> • Depression and anxiety (HADS) <br> • Caregiving stress (RSS) <br> • Service utilisation (CSRI) | £2,586 per QoL-AD point | Low/Absent | High ICER for the QoL-AD. Carers of PwDs within the intervention group reported a significant increase in anxiety on a subscale of the GHQ-28. No cost-effectiveness in terms of QALY gains: intervention more costly; negligible difference in QALYs between intervention and control arms for both participants with dementia and carers. | Medium |
| 9 | Mervin et al. 2018 [129] | Plushie robot (PARO) vs normal plushie and vs Usual Care (Nursing Home or other residential facility) | Australia | RCT CEA (10 weeks) | 415 / 0 (138 / 140 / 137) | Health and social care system | • Agitation (CMAI-SF) <br> • Medication use (dementia drugs, antidepressants, antipsychotics, opioids) | — | AU$13.01 per CMAI-SF point averted (PARO); AU$12.85 per CMAI-SF point averted (plushie) | Low/Absent | No evidence of substantial cost-effectiveness for the use of the robotic plushie compared with inexpensive normal plush toys. There were no significant differences in the average number of medications between study groups. | Low |

ADAS-Cog: Alzheimer's Disease Assessment Scale-Cognition subscale; ADCS-ADL: Alzheimer's Disease Cooperative Study-Activities of Daily Living-Activities of Daily living; AMI (E): Autobiographical memory interview (extended version); AMPS: assessment of motor and process skills-process scale; BADLS: Bristol Activities of Daily Living Scale; BGSI: Bangor Goal-Setting Interview; CMAI-SF = Cohen-Mansfield Agitation Inventory-Short Form; CSDD: Cornell Scale for Depression in Dementia; CSRI: Client Service Receipt Inventory; CT-CNLTC: Criterion Time for Certification of Needed Long-Term Care; CUA: Cost-utility analysis; DEMQOL: Dementia Quality of Life score; DEMQOL Proxy: Dementia Quality of Life score reported by a carer; EQ-5D: EuroQol-5 Dimensions; FAB: Frontal Assessment Battery; GDS: Global Deterioration Scale; GSES: Generalized Self-Efficacy Scale; GHQ-28: General Health Questionnaire—28 item version; HADS: Hospital Anxiety and Depression Scale; IDDD: interview of deterioration in daily activities in dementia-performance scale, measures need for assistance; MMSE: Mini-Mental State Examination; NPI: Neuropsychiatric Inventory; PMS/IADL: Physical Self-Maintenance Scale/Instrumental Activity of Daily Living; QALYs: Quality Adjusted Life Years; QCPR: Quality of Caregiver–Patient Relationship; QoL-AD: Quality of Life-Alzheimer's Disease scale; RAID: Rating Anxiety in Dementia; RCT: Randomised controlled trial; RS-14: Resilience Scale-14 items; RSS: Relatives' Stress Scale; SCQ: Sense of competence questionnaire; SF-12: Short Form questionnaire-12 items; WHOQoL-BREF: World Health Organization's Quality of Life Instrument–brief version

Table 5. Main characteristics of studies evaluating the cost-effectiveness of physical activity interventions.

| No. | Study | Intervention description and comparator (Setting) | Country | Type of study and economic evaluation (Time horizon) | Sample size No. PwDs/No. Caregivers (Intervention Group/Control Group) | Perspective | PwD outcome measures | Caregiver outcome measures | Incremental Cost-Effectiveness Ratio (ICER) and other cost-effectiveness measures | Cost-effectiveness assessment | Cost-effectiveness rationale | Quality of the study |
|---|---|---|---|---|---|---|---|---|---|---|---|---|
| 1 | Eckert et al. 2021 [126] | Individually-tailored exercise program vs unspecified flexibility training (Community-based: home) | Germany | RCT CEA+CUA (24 weeks) | 118 / 0 (63 / 55) | Societal | • Physical performance (SPPB) <br>• Quality of life (EQ-5D) <br>• Cognition (MMSE) <br>• Comorbidities <br>• Health Care Service utilisation | — | 92% probability of positive net monetary benefit for a WTP of €500 per point on the SPPB; 90% probability of cost-utility for a WTP of €20,000 per QALY (no QALY gains but lower healthcare costs in the intervention group) | Moderate | Partial cost-effectiveness: high probability of cost-effectiveness in terms of improved physical performance in geriatric patients with cognitive impairment following discharge from ward rehabilitation, but not in terms of improved quality of life (the exercise intervention did not achieve gains in QALYs compared to control condition). | High |
| 2 | D'Amico et al. 2015 [135] | Physical exercise regimen (walking) for patient-caregiver dyads vs Usual Care (Community-based: home) | UK | RCT CEA+CUA (12 weeks) | 52 / 52 (30 / 22) | Health and social care system Societal | • Behavioural and psychological symptoms (NPI) <br>• General health (GHQ) <br>• Quality of life (DEMQOL Proxy) <br>• Service utilisation (CSRI) | • Caregiver burden (ZBI) | £421 per NPI point, £286,440 per QALY gained (point estimates with societal perspective) | Moderate | Partial cost-effectiveness: exercise intervention is significantly cost-effective in terms of improvements in behavioural and psychological symptoms (NPI score), but the authors observe that there is no established cost-effectiveness benchmark for the NPI. Intervention is not cost-effective when considering additional cost of QALY gains. | Medium |

(Continued)

Table 5. (Continued)

| No. | Study | Intervention description and comparator (Setting) | Country | Type of study and economic evaluation (Time horizon) | Sample size No. PwDs/No. Caregivers (Intervention Group/Control Group) | Perspective | PwD outcome measures | Caregiver outcome measures | Incremental Cost-Effectiveness Ratio (ICER) and other cost-effectiveness measures | Cost-effectiveness assessment | Cost-effectiveness rationale | Quality of the study |
|---|---|---|---|---|---|---|---|---|---|---|---|---|
| 3 | Khan et al. 2018 [125] | Structured physical exercise (aerobic and resistance training at moderate-to-hard intensity) vs Usual Care (Community-based: home) | UK | RCT CUA (12 months) | 494 / 494 (329 / 165) | Health and social care system Societal | • Cognitive impairment (ADAS-Cog) • ADLs (BADLS) • Quality of life (EQ-5D, QoL-AD) • Behavioral and psychological symptoms (NPI) • Service utilisation (CSRI) • Falls and fractures | • Caregiver burden (ZBI) | Mean ICER negative: -£74,227 per QALY gained (intervention more costly and less effective) | Low/Absent | Dominated (higher costs and worse outcome). The probability that the exercise intervention is cost effective is < 1% for a WTP between £15,000 and £30,000 for an additional QALY. Patients became physically fitter due to exercise but these benefits did not translate into improvements in important cognitive outcomes. | High |

ADAS-Cog: Alzheimer's Disease Assessment Scale-Cognition subscale; ADLs: Activities of Daily Living Scale; BADLS: Bristol Activities of Daily Living Scale; CSRI: Client Service Receipt Inventory; CUA: Cost-utility analysis; DEMQOL Proxy: Dementia Quality of Life score reported by a carer; EQ-5D: EuroQol-5 Dimensions; GHQ: General Health Questionnaire; MMSE: Mini-Mental State Examination; NPI: Neuropsychiatric Inventory; QALYs: Quality Adjusted Life Years; QoL-AD: Quality of Life-Alzheimer's Disease scale; RCT: Randomised controlled trial; SPPB: Short Physical Performance Battery; ZBI: Zarit Burden Interview (self-reported questionnaire used to assess carer burden).

**Table 6. Main characteristics of studies evaluating the cost-effectiveness of indirect strategies.**

| No. | Study | Intervention description and comparator (Setting) | Country | Type of study and economic evaluation (Time horizon) | Sample size No. PwDs/No. Caregivers (Intervention Group/Control Group) | Perspective | PwD outcome measures | Caregiver outcome measures | Incremental Cost-Effectiveness Ratio (ICER) and other cost-effectiveness measures | Cost-effective. Assess. | Cost-effectiveness rationale | Quality of the study |
|---|---|---|---|---|---|---|---|---|---|---|---|---|
| 1.1 | Michalowsky et al. 2019 [63] | Dementia Care Management (Delphi-MV trial) vs Usual Care (Community-based: home) | Germany | RCT CUA (24 months) | 444 / 0 (315 / 129) | Health and social care system Societal | • Health related Quality of life (SF-12) • Cognitive impairment (MMSE) • Depression and anxiety (GDS) • B-ADL • Comorbidity (CCI) • Service utilisation (CSRI) • Time to institutionalisation | — | In the base-case analysis, Incremental cost per QALY < 0 | High | In the base-case analysis, DCM dominated the usual care in PwDs living alone while the ICER of the DCM for those living with a caregiver valued €26,851 per QALY (below the NICE's threshold of £30,000 per QALY). The probability of the DCM being cost-effective is 56% at €0 WTP and increases to 88% at a WTP of € 40,000 per QALY (close to the NICE's threshold). | High |
| 1.2 | Rädke et al. 2020 [116] | Dementia Care Management (Delphi-MV trial) vs Usual Care (Community-based: home) | Germany | RCT CUA (24 months) | 444 / 0 (315 / 129) | Health and social care system | • Health related Quality of life (SF-12) • Cognitive impairment (MMSE) • Depression and anxiety (GDS) • B-ADL • Comorbidity (CCI) • Service utilisation (CSRI) | — | In the base-case analysis, Incremental cost per QALY < 0 | High | DCM dominated usual care in PwDs >80, female, living alone, with functional impairment (B-ADL), with cognitive deficit (MMSE). The probability of the DCM being cost-effective at a WTP of € 40,000 per QALY (close to the NICE's threshold of £30,000 per QALY) was higher in females (96% versus 16% for males), in those living alone (96% versus 26% for those living not alone), in those being moderately to severely cognitively (100% versus 3% for patients without cognitive impairment) and functionally impaired (97% versus 16% for patients without functional impairment), and in PwDs having a high comorbidity (96% versus 26% for patients with a low comorbidity). | High |
| 2 | Wimo et al. 1995 [67] | Group living for dementia patients vs Home living and Institutional living (Group living) | Sweden | Prospective study with Markov model (Expected life-length of 8 years) | 108 / 0 (46 / 39 home; 23 instit.) | Health and social care system Societal | • Degree of dementia (GDS) • QALYs gained (IWB scale) | — | Incremental cost per QALY gained < 0 (compared to both institutionalisation and living at home) | High | Dominant (better outcomes and lower costs) even at a low WTP. Additional evidence needed since the study was not a RCT. | Medium |
| 3 | MacNeil Vroomen et al. 2016 [62] | Case management (Intensive Case Management Model; Linkage Model) vs Usual Care (Community-based: home) | Netherlands | Prospective study CEA+CUA (24 months) | 521 / 521 (234 ICMM; 214 LM / 73 control) | Societal | • Behavioural and psychological symptoms (NPI) • Quality of life (EQ-5D) | • Mental health (GHQ) • Quality of life (EQ-5D) | Mean ICERs: €9,581,433 per QALY (ICMM vs control); €2,236,139 per QALY (LM vs control) (combined QALYs for patient and caregiver). The loss of one combined QALY is associated with cost-saving. | Moderate | For all outcomes (NPI, GHQ, QALYs), the probability that the ICMM was cost-effective in comparison with LM and the control group is larger than 97% at a WTP of 0 €/incremental unit of effect. However, cost savings were accompanied by a small (non-significant) negative effect on QALYs for the PwDs in both ICMM and LM groups compared to the control group. Additional evidence needed since the study was not a RCT. | Medium |
| 4 | Wimo et al. 1994 [70] | Adult Day Care vs Wait-list (Community-based: Day Care) | Sweden | Prospective study CEA (12 months) | 100 /0 (55 / 45) | Health and social care system | • Quality of life (IWB, Rosser index) • Cognitive impairment (MMSE) • ADLs and Behaviour (MDDAS) | — | Incremental cost per unit of effectiveness < 0 | Moderate | Day Care was both cost-saving and had better outcomes. Since the changes between the groups were not significant regarding the cost-effectiveness quotient, the authors could not conclude that day care was cost-effective. However, for a subgroup of patients with the most distressed psychosocial situations, day care has shown to be cost-effective (it provides the same QoL indices of the comparator but at a lower cost). | Medium |
| 5 | Melis et al. 2008 [120] | Dutch Geriatric Intervention Programme (preventive nurse visits) vs Usual Care (Community-based: home) | Netherlands | RCT CEA (6 months) | 151 / 0 (85 / 66) | Health and social care system | • IADLs (GARS-3) • Mental well-being (MOS-20MH) | — | Mean ICER of €3,418 per successful treatment (prevented functional decline accompanied by improved well-being) (point estimate) | Moderate | Partial cost-effectiveness. Dominant intervention with a probability of 34.6%. Cost-effectiveness with a probability of 95% for a WTP of €34,000 for a successful treatment (no established WTP thresholds for unit of effectiveness). | Medium |

(*Continued*)

**Table 6.** (Continued)

| No. | Study | Intervention description and comparator (Setting) | Country | Type of study and economic evaluation (Time horizon) | Sample size No. PwDs/No. Caregivers (Intervention Group/ Control Group) | Perspective | PwD outcome measures | Caregiver outcome measures | Incremental Cost-Effectiveness Ratio (ICER) and other cost-effectiveness measures | Cost-effective. Assess. | Cost-effectiveness rationale | Quality of the study |
|---|---|---|---|---|---|---|---|---|---|---|---|---|
| 6 | Livingston et al. 2019 [65] | MARQUE intervention (mandatory training sessions for staff and implement new procedures to reduce agitation) vs Usual Care (Nursing Home) | UK | RCT CUA (8 months) | 318 / 354 (PwD: 155/163) (Staff: 175/179) | Health and social care system | • Agitation (CMAI) • Behavioural and psychological symptoms (NPI) • Dementia severity (CDR) • Antipsychotic drug use • Quality of life (DEMQOL-Proxy, EQ-5D) • Service utilisation (CSRI) | • Caregiver burnout (MBI) • Sense of competence (SCD) • Abusive behaviour by staff (STS) | Mean ICER of £14,064 per QALY gained (patient) | Low/ Absent | The MARQUE intervention was not found to be significantly less costly than usual care while it was not effective for reducing agitation and antipsychotic drug consumption or increasing QALYs. Very low probability of cost-effectiveness the mean incremental cost per QALY gained (£14,064) is less than the NICE threshold of £20,000 per QALY, but with a relatively low probability (62%). | Medium |
| 7 | Meeuwsen et al. 2013 [66] | Memory clinics (providing drugs and non-pharmacological interventions) vs Care by GP (Community-based: Memory clinics) | Netherlands | RCT CUA (12 months) | 160 / 160 (83 / 77) | Societal | • Quality of life (EQ-5D) • ADLs • IADLs • Service utilisation | • Quality of life (EQ-5D) | Mean ICER of €41,442 per QALY (patient + caregiver). The loss of one combined QALY is associated with cost-savings | Low/ Absent | Compared to GP's care, treatment provided by the memory clinics was on average €1,024 cheaper and showed a non-significant decrease of 0.025 QALYs. There was no evidence that memory clinics were more cost-effective compared to GPs with regard to post-diagnosis treatment and coordination of care of patients with dementia in the first year after diagnosis. | High |
| 8 | Howard et al. 2021 [68] | Assistive technology and telecare for independent living vs limited control technology (Community-based: home) | UK | RCT CEA+CUA (3-6-12-24 months) | 495 / 495 (248 / 247) | Health and social care system Societal | • Time to residential care • Number of adverse events • Quality of life (EQ-5D) • Cognitive impairment (MMSE) • Activities of daily living (BADLS) | • Caregiver burden (ZBI) • Depressions (CES-D) • Anxiety (STAI) | Mean ICER assessed after 24 months under the societal perspective £ 33,672 per QALY (participant) | Low/ Absent | Time living independently outside a care home was not significantly longer in participants. Participants attained fewer QALYs at non-significantly lower costs than controls | High |
| 9 | Van de Ven et al. 2014 [130] | Dementia Care Mapping V s Usual care (Nursing Home) | Netherlands | RCT CCA (18 months) | 318 / 319 (PwD: 154/164) (Staff: 141/178) | Health and social care system | • Health Care services utilisation • Psychotropic drug use • Falls and fractures | • Absenteeism (nursing home staff) | Intervention is cost-neutral compared to usual care without significant positive effects on outcomes | Low/ Absent | Cost-neutral intervention without significant improvements in outcomes. The intervention group showed lower costs associated with outpatient hospital appointments over time than the control group but these costs are negligible compared to the costs associated with daily care. Besides, the average number of falls and the use of psychotropic drugs were not significantly different between the intervention group and the control group. | Low |
| 10 | Meads et al. 2019 [133] | Dementia Care Mapping V s Usual Care (Nursing Home) | UK | RCT CEA+CUA (16 months) | 726 / 0 (418/308) | Health and social care system | • Agitation (CMAI) • Health outcomes (FAST) • Dementia (CDR) • Health care use • Quality of life (EQ-5D, DEMQOL-proxy) | — | In the base-case analysis: £64,380 per QALY; £272 per CMAI unit improvement. | Low/ Absent | Costs higher in the intervention arm than in the control arm, and small QALY gains. The base-case estimate of the cost of CMAI unit improvement (£272) is higher than previous estimates. | High |

ADLs: Activities of Daily living; BADLS: Bristol Activities of Daily Living Scale; B-ADL: Bayer-Activities of Daily Living Scale; CCA: Cost-consequence analysis; CCI: Charlson Comorbidity Index; CDR: Clinical Dementia Rating; CES-D: Center for Epidemiologic Studies Depression Scale; CMAI = Cohen- Mansfield Agitation Inventory; CSRI: Client Service Receipt Inventory; CUA: Cost-utility analysis; DEMQOL Proxy: Dementia Quality of Life score reported by a carer; EQ-5D: EuroQol-5 Dimensions; FAST: Functional Assessment Staging Test; GARS-3: Groningen Activity Restriction Scale-3; GDS: Global Deterioration Scale; GHQ: General Health Questionnaire; IADLs: Instrumental Activities of Daily Living; IWB: Index of well-being; MBI: Maslach Burnout Inventory; MDDAS: Multi-Dimensional Dementia Assessment Scale; MMSE: Mini-Mental State Examination; MOS-20MH: mental health subscale of the Medical Outcomes Study Short Form; NPI: Neuropsychiatric Inventory; QALYs: Quality Adjusted Life Years; QUIS: Quality of Interactions Scale; RCT: Randomised controlled trial; SCD: Sense of Competence in Dementia; SF-12: Short Form questionnaire-12 items; STAI: State-Trait Anxiety Inventory; STS: Staff Tactics Scale; ZBI: Zarit Burden Interview (self-reported questionnaire used to assess carer burden).

**Table 7. Main characteristics of studies evaluating the cost-effectiveness of interventions primarily aimed at supporting family caregivers.**

| No. | Study | Intervention description and comparator (Setting) | Country | Type of study and economic evaluation (Time horizon) | Sample size No. PwDs/No. Caregivers (Intervention Group/Control Group) | Perspective | PwD outcome measures | Caregiver outcome measures | Incremental Cost-Effectiveness Ratio (ICER) and other cost-effectiveness measures | Cost-effective. Assess. | Cost-effectiveness rationale | Quality of the study |
|---|---|---|---|---|---|---|---|---|---|---|---|---|
| 1.1 | Knapp et al. 2013 [71] | Individual therapy program for informal caregivers (START) vs Usual Care alone (Community-based: mental health and neurological outpatient dementia services) | UK | RCT CEA+CUA (8 months) | 260 / 260 (173 / 87) | Health and social care system | • Behavioural and psychological symptoms (NPI) | • Depression and anxiety (HADS) • Quality of life (EQ-5D) • Caregiver burden (ZBI) • Coping strategies (COPE) • Health and social care use | In the base-case analysis (only caregiver's costs): £6,000 per QALY gained (caregiver); £118 per HADS point (caregiver) | High | The short-term intervention had a 99% probability of being cost-effective for carers at the NICE's WTP threshold of £30,000 per QALY gained. Moreover, START showed a high probability of cost-effectiveness on the HADS-T (Hospital Anxiety and Depression Scale) measure even though the authors were not aware of societal WTP for gauging cost-effectiveness on the HADS scale. | High |
| 1.2 | Livingston et al. 2014 [117] | Individual therapy program for informal caregivers (START) vs Usual Care alone (Community-based: mental health and neurological outpatient dementia services) | UK | RCT CEA+CUA (24 months) | 209 / 209 (140 / 69) | Health and social care system | • Dementia severity (CDR) • Quality of life (QoL-AD) • Behavioural and psychological symptoms (NPI) • Service utilisation (CSRI) | • Depression and anxiety (HADS) • Quality of life (EQ-5D) • Caregiver burden (ZBI) • Coping strategies (COPE) • Service utilisation (CSRI) | In the base-case analysis: 1) considering carer-and- patient costs combined, Incremental cost per unit of outcome < 0; 2) considering carer-only costs: £244 per QoL-AD point (patient) £12,400 per QALY gained (caregiver) £179 per HADS point (caregiver) | High | 1) Considering carer-and- patient costs combined, START dominates usual care when looking at carer outcomes, total HADS score and QALYs (outcomes are better and costs not significantly different) and the intervention had a 70% probability of being cost-effective in terms of carer QALY gain at the NICE's WTP threshold of £30,000 per QALY. 2) Considering carer-only costs, cost per carer QALY is less than the lower NICE threshold with a 75% likelihood of cost-effectiveness at the NICE's WTP threshold of £30,000 per QALY. | Medium |

(*Continued*)

Table 7. (Continued)

| No. | Study | Intervention description and comparator (Setting) | Country | Type of study and economic evaluation (Time horizon) | Sample size No. PwDs/No. Caregivers (Intervention Group/Control Group) | Perspective | PwD outcome measures | Caregiver outcome measures | Incremental Cost-Effectiveness Ratio (ICER) and other cost-effectiveness measures | Cost-effective. Assess. | Cost-effectiveness rationale | Quality of the study |
|---|---|---|---|---|---|---|---|---|---|---|---|---|
| 1.3 | Livingston et al. 2019 [118] | Individual therapy program for informal caregivers (START) vs Usual Care alone (Community-based: mental health and neurological outpatient dementia services) | UK | RCT CEA (6 years follow–up) | 222 / 222 (150 / 72) | Health and social care system | • Behavioural and psychological symptoms (NPI) • Service utilisation (CSRI) | • Depression and anxiety (HADS) • Caregiver burden (ZBI) • Service utilisation (CSRI) | Intervention is cost-saving compared to usual care but has positive effects on outcomes (e.g. mean difference in HADS scores of -2.00 points) | High | The positive difference in outcomes is small but statistically significant, greater than the minimally clinically important difference and is sustained after 6 years. The difference in costs appears to be economically large (e.g. cost per patient in the intervention group is around a third of the cost in the control group) although for PwDs there was no significant difference in time to care home admission or death. | Medium |
| 2 | Nichols et al. 2008 [73] | Psychosocial intervention for informal caregivers (REACH II) Vs Usual care (Community-based: home) | USA | RCT CEA (6 months) | 112 / 112 (55 / 57) | Societal | • Cognitive impairment (MMSE) • Behavioural and psychological symptoms (RMBPC) • ADLs (Katz scale) • IADLs (Lawton Brody scale) • Service utilisation | • Time spent caregiving • Caregiver bother (RMBPC) • Depression (CES-D) • Service utilisation • Social support | $4.96 per hour not spent in caregiving (the cost of an additional hour of non-caregiving time that could be "purchased" by the intervention) | High | Intervention was cost-effective if one was willing to spend $4.96 per day for 1 extra hour of non-caregiving time for each family caregiver. Moreover, the intervention could be thought of as being financially positive because it resulted in $10.56 ($8.12 of caregiver hourly wage × 1.3 hours) of time gained versus $4.96 of intervention cost per hour per day per caregiver. | Medium |
| 3 | Drummond et al. 1991 [69] | Caregiver support program (nurse visits, support groups and respite care) vs Usual Care (conventional community nursing care) (Community-based: home) | Canada | RCT CUA (6 months) | 0 / 42 (22 / 20) | Health and social care system | — | • Depression (CES-D) • Anxiety (STAI) • Quality of life (CQLI) | Mean ICER: CA$20,036 per QALY gained | Moderate | Incremental cost per QALY gained compares favourably with other health care interventions. However, evidence of cost-effectiveness was considered limited due to the statistically non-significant difference in outcome levels. Further larger studies are required. | Low |

(Continued)

**Table 7.** (Continued)

| No. | Study | Intervention description and comparator (Setting) | Country | Type of study and economic evaluation (Time horizon) | Sample size No. PwDs/No. Caregivers (Intervention Group/Control Group) | Perspective | PwD outcome measures | Caregiver outcome measures | Incremental Cost-Effectiveness Ratio (ICER) and other cost-effectiveness measures | Cost-effective. Assess. | Cost-effectiveness rationale | Quality of the study |
|---|---|---|---|---|---|---|---|---|---|---|---|---|
| 4 | Shaw et al. 2020 [76] | FamTechCare telehealth intervention to assist caregivers vs Telephone intervention (Community-based: home) | USA | RCT CEA (3 months) | 56 / 68 (31 / 37) | Health and social care system | — | • Caregiver depression (CES-D) • Caregiver competence (SSCQ) | Mean ICERs: $222.17 (per dyad) for 1-point improvement in CES-D score (depression); $436.53 (per dyad) for 1-point improvement in SSCQ score (competence) | Moderate | Partial cost-effectiveness: a caregiver's WTP amount on improvement in SSCQ score (based on a different trial focused on training caregivers) is used as a threshold to determine the cost-effectiveness of the intervention. However, the authors recognise that established external WTP thresholds for the considered units of effectiveness do not exist. | Low |
| 5 | Gaugler et al. 2003 [131] | Adult day care service to support informal caregivers vs Usual Care (Community-based: Day Care) | USA | Prospective study CEA (3 months; 1 year) | 0 /201 (80/121) | Societal | • Behaviour Problem Scale • ADL | • Stress (ROS) • Depression (CES-D) | Mean ICERs calculated as the cost necessary to alleviate role overload and depression by one unit: 1) $6.83/day per unit of ROS score; $2.90/day per unit of CES-D score (over 3-months period); 2) $4.51/day per unit of ROS score; $2.20/day per unit of CES-D score (over 1-year period) | Moderate | Partial cost-effectiveness: the daily costs of carer's benefits were reduced over a 1-year period. Long-term utilization helped to lessen the time carers spent managing symptoms associated with dementia (i.e., ADL dependencies and behaviour problems) and allowed caregivers to spend more time in work-related activities. No established external WTP thresholds for unit of effectiveness. | Low |

*(Continued)*

Table 7. (Continued)

| No. | Study | Intervention description and comparator (Setting) | Country | Type of study and economic evaluation (Time horizon) | Sample size No. PwDs/No. Caregivers (Intervention Group/Control Group) | Perspective | PwD outcome measures | Caregiver outcome measures | Incremental Cost-Effectiveness Ratio (ICER) and other cost-effectiveness measures | Cost-effective. Assess. | Cost-effectiveness rationale | Quality of the study |
|---|---|---|---|---|---|---|---|---|---|---|---|---|
| 6 | Joling et al. 2013 [72] | Family meetings for informal caregivers vs Usual Care (Community-based: home) | Netherlands | RCT CUA (12 months) | 192 / 192 (96 / 96) | Societal | • Health Related Quality of life (SF-12) • Service utilisation (hospital and long-term care facilities) | • Health Related Quality of life (SF-12) • Depression and anxiety (MINI) • Service utilisation • Work absenteeism | Mean ICERs: −€807,703 per QALY (dyad: carer +patient), -240,247 per QALY (patient), −€24,472 per QALY (caregiver) [intervention more costly and less effective] | Low/ Absent | The intervention is not considered cost-effective. Since the differences in effects on all outcomes were very small, this resulted in very large ICERs that are very sensitive to uncertainty in incremental effect. The probability that the intervention was considered cost-effective was 36% for the outcome QALY per dyad (patient+carer) when the ceiling ratio is set at €30,000/QALY). For caregivers separately this probability was 85% for a ceiling ratio of €30,000/QALY. For patients this probability was around 29% for a ceiling ratio of €30,000/ QALY. | High |
| 7 | Wilson et al. 2009 [132] | Social care intervention for informal caregivers (contact with a befriender facilitator) vs Usual Care (Community-based: home) | UK | RCT CUA (15 months) | 0 / 190 (93 / 97) | Societal | — | • Depression and anxiety (HADS) • Quality of life (EQ-5D) | Mean ICERs in the base-case: £105,954 per QALY (caregiver) £28,848 per QALY (carer+patient) | Low/ Absent | It is unlikely that befriending is a cost-effective intervention. The intervention had only a 42.2% probability of being cost-effective in terms of carer QALY gain at the NICE's WTP threshold of £30,000 per QALY. The intervention had only a 51.4% probability of being cost-effective in terms of dyad (carer+patient) QALY gain at the NICE's WTP threshold of £30,000 per QALY. | High |

ADLs: Activities of Daily living; CES-D: Center for Epidemiologic Studies Depression Scale; COPE: self-completed measure of carer coping strategies; CQLI: Caregiver Quality of Life Instrument; CSRI: Client Service Receipt Inventory; CUA: Cost-utility analysis; EQ-5D: EuroQol-5 Dimensions; HADS: Hospital Anxiety and Depression Scale; IADLs: Instrumental Activities of Daily Living; MMSE: Mini-Mental State Examination; MINI: Mini International Neuropsychiatric Interview; NPI: Neuropsychiatric Inventory; QALYs: Quality Adjusted Life Years; RCT: Randomised controlled trial; RMBPC: Revised Memory and Behavior Problem Checklist; ROS: Role Overload Scale; SF-12: Short Form questionnaire-12 items; SSCQ: Short Sense of Competence Questionnaire; STAI: State-Trait Anxiety Inventory; ZBI: Zarit Burden Interview (self-reported questionnaire used to assess carer burden).

**Table 8. Main characteristics of studies evaluating the cost-effectiveness of multicomponent interventions.**

| No. | Study | Intervention description and comparator (Setting) | Country | Type of study and economic evaluation (Time horizon) | Sample size No. PwDs/No. Caregivers (Intervention Group/Control Group) | Perspective | PwD outcome measures | Caregiver outcome measures | Incremental Cost-Effectiveness Ratio (ICER) and other cost-effectiveness measures | Cost-effective. Assess. | Cost-effectiveness rationale | Quality of the study |
|---|---|---|---|---|---|---|---|---|---|---|---|---|
| 1.1 | Ballard et al. 2018 [77] | WHELD intervention (person-centred care, management of agitation, physical exercise and psychosocial approaches) vs Usual Care alone (Nursing Home) | UK | RCT CCA (9 months) | 553 / 0 (257 /296) | Health and social care system | • Quality of life (DEMQOL Proxy) • Dementia severity (CDR, FAST) • Agitation (CMAI) • Behavioural and psychological symptoms (NPI) • Mood (CSSD) • Antipsychotic drug use • Quality of interaction (QUIS) • Service utilisation (CSRI) | — | In the base-case analysis, incremental cost per unit of effectiveness < 0 | High | The intervention was dominant (better outcomes and lower costs than the comparator). WHELD intervention confers benefits in terms of QoL (DEMQOL Proxy), agitation (CMAI score), and neuropsychiatric symptoms (NPI score), albeit with relatively small effect sizes in terms of clinically significant benefits (on CMAI and NPI). However, the benefits to the broader population of people with dementia in care homes make this a meaningful benefit in the quality of care. No significant reduction in antipsychotic use was achieved, and antipsychotic use was stable in both study groups. | Medium |
| 1.2 | Romeo et al. 2018 [78] | WHELD intervention (person-centred care, management of agitation, physical exercise and psychosocial approaches) + Usual care vs Usual Care alone (Nursing Home) | UK | RCT CEA+CUA (9 months) | 549 / 0 (267 / 282) | Health and social care system | • Agitation (CMAI) • Quality of life (DEMQOL Proxy) • Service utilisation (CSRI) • Antipsychotic drug use | — | Mean ICERs in the base case analysis: -£137.978 per QALY gained -£348 per point improvement in agitation (CMAI score) | High | The intervention was dominant (better outcomes and lower costs than the comparator) for a wide range of societal WTP thresholds. The assessment of cost-effectiveness and parameter uncertainty confirmed that the intervention would have a high probability of being cost effective. If decision makers were willing to pay £30,000 for QALY gained (£200 for each point improvement in CMAI score), the probability that the intervention is cost effective is as high as 90% (100%). | Medium |
| 2 | Steinbeisser et al. 2020 [83] | MAKS intervention vs Usual Care (Community setting: day care centers) | Germany | RCT CEA (6 months) | 453 / 0 (263 / 190) | Societal | • Cognition (MMSE) • ADLs (ETAM) • Service utilisation | — | In the base-case analysis, incremental cost per unit of effectiveness < 0 | High | The intervention has a high probability to be dominant (better outcomes and lower costs than the comparator). It has: 78% (95%) probability of cost-effectiveness for a WTP of €0 (€939.66) for 1 MMSE point; 77.4% (95%) probability of cost-effectiveness for a WTP of €0 (€ 937.73) for 1 ETAM point. For outcome measures such as MMSE and ETAM scores, no societal WTP thresholds have been defined. | High |
| 3 | Wolfs et al. 2011 [79] | Integrated approach (map of the patient and caregiver needs to develop a personalised treatment course) Vs Usual Care (Community-based: Diagnostic research centre for psycho-geriatrics) | Netherlands | RCT CUA (1 year) | 219 / 0 (131 / 88) | Societal | • Quality of life (EQ-5D) • Cognitive impairment (MMSE) • Behavioural and psychological symptoms (NPI) • IADLs (Lawton Brody scale) • Depression (CSDD) • Service utilisation | — | Mean ICER: €1,267 per QALY gained | Moderate | Partial cost-effectiveness: the intervention was cost-effective in terms of QALYs for ambulatory PwDs but not in terms of improvements in clinical measures such as cognitive impairment or behavioural and psychological symptoms (due to relevant statistical uncertainty). | Low |

(*Continued*)

**Table 8.** (Continued)

| No. | Study | Intervention description and comparator (Setting) | Country | Type of study and economic evaluation (Time horizon) | Sample size No. PwDs/No. Caregivers (Intervention Group/Control Group) | Perspective | PwD outcome measures | Caregiver outcome measures | Incremental Cost-Effectiveness Ratio (ICER) and other cost-effectiveness measures | Cost-effective. Assess. | Cost-effectiveness rationale | Quality of the study |
|---|---|---|---|---|---|---|---|---|---|---|---|---|
| 4 | El Alili et al. 2020 [84] | Namaste Care Family Program vs Usual Care (Nursing home) | Netherlands | RCT CEA+CUA (12 months) | 231 / 116 (116 / 115) | Health care system Societal | • Quality of life (EQ-5D, QUALID) • Service utilisation | • Caregiving (GAIN) • Loss of productivity | Mean ICERs in the main analysis (societal perspective): -€8,919 for 1 point improvement/reduction in QUALID score; €7,310 for 1 point improvement in GAIN score; -€315,671 per QALY gained | Moderate | The intervention was potentially dominant (better outcomes and lower costs than the comparator) but there is statistical uncertainty surrounding the results: the probability of cost-effectiveness did not exceed 70% for any threshold value of WTP for one additional QALY. Moreover, for outcome measures such as the QUALID and GAIN, no societal WTP thresholds have been defined. | Medium |
| 5 | Sogaard et al. 2014 [80] | Psychosocial intervention (DAISY) vs Usual Care (Community-based: Primary care and memory clinics) | Denmark | RCT CUA (36 months) | 330 / 330 (163 / 167) | Health and social care system Societal | • Quality of life (EQ-5D) • Service utilisation • Institutionalisation rates | • Quality of life (EQ-5D) • Time spent caregiving (RUD) | Mean Incremental cost per QALY <0 (additional average cost of €3,401; difference in mean QALYs: -0.09) | Low/ Absent | The intervention was more costly and less effective even though the authors found no significant difference in both the measured costs and QALYs between the intervention and control groups. The probability of cost-effectiveness from a societal perspective did not exceed 36% for any threshold value of WTP for one additional QALY. The alternative scenario analysis showed that the probability of cost-effectiveness could increase if the cost perspective were restricted to formal health care and if the programme were focused only on patients and caregivers with special needs. | High |
| 6 | Eloniemi-Sulkava et al. 2009 [81] | Multicomponent support intervention for couples vs Usual Care (Community-based) | Finland | RCT CCA (2 years) | 125 / 125 (63 / 62) | Health and social care system | • Comorbidity (CCI) • Physical functioning (Barthel Index) • Behavioural and psychological symptoms (NPI) • Service utilisation • Institutionalisation | • Caregiver burden (ZBI) | A decrease in healthcare costs for the intervention group (the mean difference was €7,985 per capita per year) due to a reduction in the use of community services and expenditures (difference not considering the intervention costs) | Low/ Absent | The authors found a substantial equivalence in the institutionalisation risk between the control and the treated groups and lower healthcare costs for the intervention group. However, when the intervention costs were included, the differences between the groups were not significant. | Low |

CCA: Cost-consequence analysis; CCI: Charlson Comorbidity Index; CDR: Clinical Dementia Rating; CMAI = Cohen- Mansfield Agitation Inventory; CSDD: Cornell Scale for Depression in Dementia; CSRi: Client Service Receipt Inventory; CUA: Cost-utility analysis; DEMQOL Proxy: Dementia Quality of Life score reported by a carer; EQ-5D: EuroQol-5 Dimensions; ETAM: Erlangen Test of Activities of Daily Living in Persons with Mild Dementia or Mild Cognitive Impairment; FAST: Functional Assessment Staging Test; GAIN: Gain in Alzheimer Care Instrument for family caregivers; IADLs: Instrumental Activities of Daily Living; MMSE: Mini-Mental State Examination; NPI: Neuropsychiatric Inventory; QUALID: Quality of Life in Late-Stage Dementia; QALYs: Quality Adjusted Life Years; QUIS: Quality of Interactions Scale; RCT: Randomised controlled trial; RUD: Resource utilization in dementia-instrument; ZBI: Zarit Burden Interview (self-reported questionnaire used to assess carer burden).

directed at patient-caregiver dyads, while nine studies focused on specific programmes supporting informal caregivers of community-dwelling PwDs [68, 69, 71–73, 76, 117, 131, 132].

Most studies (31 out of 39, and 28 out of 35 interventions) were conducted in European countries with comparable underlying health and social care systems (16 were based in the UK), while only seven studies were developed in other OECD countries, including four in the United States, one in Japan, one in Australia, and one in Canada.

The studies were quite heterogeneous in terms of their design, the cost items included, and the choice of outcome measures. Most studies (n = 34) used a randomised controlled trial (RCT) design, while five were non-randomised comparisons through prospective matched controlled trials [62, 67, 70, 128, 131].

Regarding the cost components considered, most of the studies–with few exceptions [76, 81, 127, 131]–identified all relevant costs for each alternative on the basis of a complete analysis of health and social care resource utilisation (medical outpatient and inpatient treatments, medications, medical aids, home care, day-care and nursing home care services, etc.). 18 of the analysed studies adopted a narrow perspective when measuring costs, looking only at health and social care, while the other 21 studies considered a broader societal perspective, including the opportunity costs of caregivers' inputs and the impacts of caring on their own health and wellbeing. In the latter studies, informal care time of family caregivers has generally been evaluated considering the average opportunity cost for lost production or leisure time and average gross wage plus non-wage labour cost (proxy good approach). Many studies have considered the variation in the use of health and social services not only as components for the calculation of costs but also as an outcome element in order to assess whether the analysed intervention was effective (compared to the comparator) in reducing the use of social and health services or in limiting the consumption of drugs. However, only five studies, focused on nursing home PwDs, have investigated the ability of SC interventions to reduce the use of psychotropic drugs [129, 130], and in particular antipsychotics [65, 77, 78].

In all the analysed studies, the SC intervention under investigation was explicitly compared—with regard to costs and outcome measures—with one or more alternatives (in most cases, the "usual care" alternative). Most studies, except for three [81, 128, 130], reported the incremental cost-effectiveness ratio (ICER) of SC interventions. When the outcome is measured in terms of utility values to account for the patient's and/or carer's QoL (e.g., using the Quality Adjusted Life Years (QALYs) gained), the cost-effectiveness analysis takes the form of a cost-utility analysis. Some studies [122, 125, 126, 128, 134, 135] calculated the net-benefits of supportive care interventions using a series of hypothetical values for the decision maker's willingness-to-pay (WTP) for an additional unit of outcome (e.g., a one-point difference in the Neuropsychiatric Inventory (NPI) score).

Twelve studies were pure cost-effectiveness analyses [70, 73, 76, 83, 118–120, 124, 127, 129, 131, 134], eleven were cost-utility analyses [63, 65–67, 69, 72, 79, 80, 116, 125, 132], twelve developed both a cost-effectiveness and a cost-utility analysis [62, 68, 71, 78, 84, 117, 121–123, 126, 133, 135], three were cost-consequence analyses [77, 81, 130], and one was a cost-benefit analysis [128].

## Quality assessment of the included studies

As indicated above, we also assessed the methodological quality of the studies included in the review using the CHEC checklist [88]. Based on the scores assigned, studies were classified as being high-, medium-, or low-quality. The quality level of the study is reported in the last column of Tables 4–8. Overall, only three studies [63, 132, 133] met all 19 criteria defined in the

checklist. Applying the CHEC criteria described in the Materials and methods section, we found 15 high-quality studies, 16 medium-quality studies, and 8 low-quality ones. Details on the ratings of the studies can be found in S1–S5 Tables.

## Evidence of cost-effectiveness of supportive care interventions from reviewed studies

In this section, we describe the results of the qualitative analysis of the studies considered in the systematic review by distinguishing between the five categories of SC strategies described in the Materials and methods section. As already mentioned, Tables 4–8 report the main characteristics of the analysed studies for each category of intervention, including: description of the intervention under evaluation and of comparator; country where the intervention was implemented; type of study, type of economic evaluation, and time horizon; sample size (i.e., number of PwDs and caregivers considered in the study as well as the size of intervention and control groups); perspective of the economic evaluation; patient outcome measures; caregiver outcome measures; mean ICER or other cost-effectiveness measures (e.g., the intervention's net benefit); assessed level of cost-effectiveness; cost-effectiveness rationale; and assessed quality of the study according to CHEC.

The analysed SC interventions are rated according to three levels of cost-effectiveness:

- high cost-effectiveness when the intervention is found to be dominant or when the incremental cost per QALY gained is below the target threshold considered by the study (often this corresponds with that currently used by the British National Institute for Health and Care Excellence-NICE [136]) with a limited amount of statistical uncertainty;

- moderate cost-effectiveness when the incremental cost per QALY gained is above but reasonably close to the target threshold, the analysis leads to mixed results (e.g., there is statistical uncertainty in the ICER value or there are no established willingness-to-pay thresholds for gains in relevant outcomes) or it is partial (e.g., there is no evidence of cost-effectiveness in terms of gains for relevant outcomes);

- low or absent cost-effectiveness when the ICER is well above the target threshold or the intervention is found to be dominated by the comparator.

**Cognitive therapy interventions.**   Table 4 summarizes the main characteristics of nine cognitive therapy interventions. One is classified as having high cost-effectiveness, six as having moderate cost-effectiveness, and two as having low or no cost-effectiveness.

Occupational therapy seems to be the most cost-effective form of cognitive therapy, as it is linked to cost savings and in most instances an improvement in patient and caregiver outcomes [119, 127]. In particular, Graff et al. [119] studied a community-based occupational therapy intervention that included both behavioural and cognitive interventions and which was directed at community-dwelling patient-caregiver dyads. The authors found average savings of approximately €1,748 per couple who had been successfully treated with the considered occupational therapy compared to usual care. Successful outcome was defined as a clinically relevant improvement in patients and caregivers for three primary outcome measures (process scale and performance scale for PwDs; competence scale for carers). The probability of occupational therapy being the dominant intervention (i.e., more effective and less costly) was estimated to be 95%. Another form of occupational therapy–the Tailored Activity Program (TAP) analysed by Gitlin et al. [127]–showed instead moderate cost-effectiveness with regards to caregiver-side outcomes (no primary outcomes for PwDs were considered).

Four cognitive stimulation therapy programmes in different settings [121–123, 134] and one learning therapy intervention for nursing home patients [128] show moderate levels of cost-effectiveness. These interventions generally highlighted potential cost-effectiveness in terms of quite heterogeneous outcome measures such as patient's self-assessed goal attainment, cognitive function (MMSE score), quality of life (QoL-AD) and needed quantity of long-term care, while they provided low or no evidence of cost-effectiveness in terms of QALY gains for PwDs or their carers (only the analysis by Orgeta et al. [123] found some evidence of cost-effectiveness in terms of caregiver's QALY gained). This makes their comparability rather complicated in terms of cost-effectiveness, even if useful considerations can still be drawn from their comparative analysis. For example, comparing the point estimates of the incremental cost for 1-point improvement in MMSE and Qol-AD scores of the maintenance cognitive stimulation therapy analysed by D'Amico et al. [122] with the CST intervention examined by Knapp et al. [134] (both were targeted towards patients with mild-to-moderate dementia in different settings), the latter would seem to be relatively more cost-effective even after correcting for the different duration of the interventions and inflation.

Finally, one reminiscence group therapy programme directed at community-dwelling patient-caregiver dyads [124] and one robotic plushie therapy intervention for institutionalised dementia patients [129] show no cost-effectiveness due to a lack of impact on outcomes, and a significant cost increase with respect to comparators.

**Physical activity interventions.** Table 5 summarizes the main characteristics of three physical therapy interventions for community-dwelling PwDs for which the evidence of cost-effectiveness is mixed: two are classified as having moderate cost-effectiveness, and one as having low or no cost-effectiveness.

Two studies on individually-tailored exercise programs report evidence of moderate cost-effectiveness compared to usual care: the interventions may be cost-effective in terms of improvements in behavioural and psychological symptoms measured with the NPI score [135] or in terms of improved physical performance [126]. However, in both cases they do not appear to be cost-effective when considering QALY gains compared to control condition. In particular, D'Amico et al. [135] observed that there was no established cost-effectiveness benchmark for the NPI with which to compare their estimates, while the estimated mean cost per QALY was rather high relative to the upper threshold (£30,000) generally associated with cost-effectiveness judgements by NICE in the UK.

On the contrary, a much larger trial, with a more extended observation period [125] provides strong evidence that a tailored, structured, moderate-to-high intensity exercise programme for PwDs in addition to usual care is unlikely to be cost-effective when compared with usual care alone: the intervention is associated with a higher cost and a lower effect (in either improvement in cognitive outcomes or QALYs) and was dominated by the comparator.

**Indirect strategies.** Table 6 summarizes the main characteristics of ten indirect interventions in different settings, described in eleven studies (two studies [63, 116] considered the same Dementia Care Management programme). Two interventions are classified as having high cost-effectiveness, three as having moderate cost-effectiveness, and five as having low or no cost-effectiveness.

Overall, group-living for certain populations of PwDs (specifically those who are on the verge of needing institutionalisation) and community-based Dementia Care Management seem to be the more cost-effective indirect strategies.

In particular, Wimo et al. [67] analysed group living (an intermediate level of care between home and institutionalisation) for dementia patients and compare it to home living and to institutional living. They found it to be dominant over both alternatives even at low WTPs. This result indicates that the intervention should be recommended as long as there are patients

suitable for group living in institutions or as long as there are patients living at home who are on the threshold of being institutionalised, even though additional evidence is needed since the study was not a RCT and was conducted several years ago.

Michalowsky et al. [63] evaluated a Dementia Care Management programme, aiming to support patients and their caregivers through coordination and management of treatment and care and consisted of a nurse-led in-depth assessment of patients' unmet needs to optimise and individualise dementia treatments (DelpHi-MV trial). The intervention was delivered in participants' homes by nurses with dementia-specific qualifications. The study was conducted from the public payer perspective, considering only outcomes for PwDs. In the base-case analysis, Michalowsky et al. [63] found the intervention to be dominant over usual care in patients living alone, and they report an ICER of €26,851 per QALY (below the NICE's threshold of £30,000 per QALY) for PwDs living with a caregiver. In particular, treated patients faced higher costs for medications but had lower costs in terms of in-hospital treatments, nursing home care, and delayed institutionalisation (the time to institutionalisation was delayed on average seven months in patients who received the intervention). The study reports cost-effectiveness with a probability 88% at a WTP of € 40,000 per QALY (close to the NICE's reference threshold of £30,000). A recent follow-up study by Rädke et al. [116] focused on subgroups of participants in the DelpHi-MV trial, and they found the intervention to be dominant over usual care for patients older than 80, females, patients living alone, and with functional impairment or a cognitive deficit; for these groups, the probability of the intervention being cost-effective at a WTP of € 40,000 per QALY was significantly higher, compared to the whole sample of patients.

Other three community-based indirect interventions [62, 70, 120] show moderate levels of cost-effectiveness. MacNeil Vroomen et al. [62] compared two forms of Dementia Care Management (Intensive Case Management Model (ICMM) and Linkage Model (LM)) with usual care, and they found for all the considered outcomes (QALYs,NPI and GHQ scores) a probability of 97% or higher for ICMM being cost-effective over LM and usual care at a WTP of €0 per incremental unit of effect. However, the interventions showed also a small negative impact on QALYs for PwDs in both case management groups compared to the control group, and so the authors observed that policy makers should decide whether this small negative effect on QALYs is acceptable based on the generated cost savings that the ICMM model appeared to provide. Moreover, they pointed out that their findings should be interpreted with caution since the study was not a randomized controlled trial. Another indirect strategy consisting in adult day-care (Wimo et al., [70]) showed moderate cost-effectiveness in a prospective study conducted several years ago: although the authors reported non-significant changes in outcomes and costs between the intervention and the control group, they noted a significant decrease in costs (while QoL measures remained at the same level) for the subgroup of patients with the highest levels of psychosocial distress. In a more recent RCT, Melis et al. [120] evaluated the cost-effectiveness of the Dutch Geriatric Intervention Programme, consisting of regular nurse visits for community-dwelling frail older people, including PwDs. The difference in the treatment effect was calculated as the difference in the proportions of successfully treated patients (prevention of functional decline together with improved wellbeing). The ICER, expressed as the total incremental cost per successful treatment, was €3,418 (-21,458 to 45,362). According to the authors, the intervention had a 95% probability of being cost-effective compared with usual care for a WTP of €34,000 for a successful treatment, but this result does not seem decisive in establishing its real value for money since there was no established WTP threshold for the outcome considered.

Lastly, three indirect interventions directed to institutionalised patients (MARQUE programme of training sessions for nursing home staff [65], two Dementia Care Mapping (DCM)

protocols [130, 133]) and two interventions for community-dwelling patients (a Dementia Care Management programme [66] and an assistive technology and telecare programme for independent living [68]) did not show evidence of cost-effectiveness.

**Interventions primarily aimed at supporting family caregivers.** Table 7 summarizes the main characteristics of seven community-based interventions to support informal caregivers of PwDs, described in nine studies (one intervention–START–has been analysed by three separate publications [71, 117, 118]). Two interventions are classified as having high cost-effectiveness, three as having moderate cost-effectiveness, and two as having low or no cost-effectiveness.

Some psychosocial interventions that target problem areas linked to informal caregivers' risks and QoL provide evidence of higher cost-effectiveness. In particular, Nichols et al. [73] considered a psychosocial intervention (REACH II) for caregivers of community-dwelling dementia patients, consisting of individual sessions and telephone-administered support group sessions. In this case, the ICER represented the cost of an additional hour of non-caregiving time that could be "purchased" by the intervention. There was no significant difference in formal healthcare use between the control and intervention dyads (carers and patients), while there was a significant reduction of hours of provided care for the caregivers in the intervention group compared to those in the control group. Nichols et al. found that the six-month intervention was cost-effective if one was willing to spend $4.96 per day for one extra hour of non-caregiving time for each caregiver.

Three studies [71, 117, 118] analysed the cost-effectiveness of the "Strategies for Relatives" (START) intervention, an individual psychosocial therapy programme which employed a similar therapeutic approach of the REACH II intervention and was aimed at informal caregivers to help them cope with the illness faced by their non-institutionalised relative. This particular approach consisted of an eight-session, manual-based coping intervention delivered by supervised psychology graduates to family carers of PwDs in addition to usual treatment. A first study by Knapp et al. [71] examined the short-term (eight months) cost-effectiveness of START, finding that the intervention had a 99% chance of being cost-effective compared with usual treatment alone at a WTP threshold of £30,000 per QALY gained (the higher threshold currently used by NICE). Livingston et al. [117] considered the START intervention for a longer timeframe of 24 months, and they found it to be dominant over usual care when looking at caregiver-side outcomes (such as the caregiver's QALYs and depression and anxiety measures) and considering carer-and-patient costs combined; moreover, the intervention had a 70% probability of being cost-effective in terms of carer QALY gain at the WTP threshold of £30,000 per QALY. A 2019 follow-up by Livingston et al. [118] found that, after six years, the positive difference in outcomes between the intervention and control groups was small but statistically significant and sustained, whereas the difference in costs was economically large, despite the fact that there was no significant difference in some patient-side outcomes such as time to institutionalisation or death. These studies also reported cost-effectiveness of START with respect to the HADS scale for depression and anxiety in caregivers (e.g., a mean ICER of £179 for 1-point reduction in the HADS total score), even though a commonly accepted reference WTP threshold for this particular outcome is unknown.

Three interventions to support family carers [69, 76, 131] are characterised by moderate cost-effectiveness. Drummond et al. [69] focused on a caregiver support program that included nurse visits, support groups and respite care, and they found limited evidence of cost-effectiveness due to the statistically non-significant difference in outcome levels between the intervention and control groups. Gaugler et al. [131] analysed an adult day care service with the explicit aim of supporting informal caregivers. The mean ICERs of the intervention were calculated as the cost necessary to alleviate role overload (ROS score) and depression

(CES-D score) by one unit both in the short and long term. The authors found that the daily costs of carers' benefits were reduced over a 1-year period (to $4.51/day per unit of ROS score and $2.20/day per unit of CES-D score, respectively) and that the long-term utilisation of day care could help to lessen the time caregivers spent managing symptoms associated with dementia and allow them to spend more time in work-related activities. Shaw et al. [76] compared a telehealth intervention to assist caregivers to traditional telephone-based assistance, and they found that the intervention could be close to the WTP threshold for an increase in caregiver competence score (measured from the Short Sense of Competence Questionnaire–SSCQ) considered for other caregiving-related interventions. In all these three cases, the lack of commonly accepted WTP thresholds for the outcomes considered reduces the possibility of measuring the real cost-effectiveness of the interventions.

Finally, one intervention revolving around regular family meetings for informal caregivers [72] and a structured befriending service for family carers [132] were not found to be cost-effective compared to usual care in terms of QALYs gained by carers or dyads.

**Multicomponent interventions.** Table 8 summarizes the main characteristics of six multicomponent interventions, described in seven studies (one intervention–WHELD–has been analysed by two separate publications [77, 78]). Two are classified as having high cost-effectiveness, two as having moderate cost-effectiveness, and two as having low or no cost-effectiveness.

A multicomponent programme for nursing home patients shows high levels of cost-effectiveness: the UK-based "Improving Wellbeing and Health for People with Dementia" (WHELD) intervention. This approach consists of a protocol to manage agitation coupled with physical exercise and psychosocial activities, all within a person-centred care framework. The protocol is focused on training care staff and promoting tailored person-centred activities and social interactions; it also involves the development of a system for triggering the appropriate review of antipsychotic medications by the prescribing physician. A preliminary study by Ballard et al. [77] found evidence of the potential high cost-effectiveness of WHELD. In particular, the intervention produced significant benefits in terms of patients' QoL measured with a Dementia Quality of Life (DEMQOL) proxy (i.e., the DEMQOL score reported by the carer), agitation (Cohen-Mansfield Agitation Inventory-CMAI score), and overall neuropsychiatric symptoms (NPI score), especially for people with moderately severe dementia. Taking into account the health and social care costs, the authors found that the WHELD intervention reduced costs compared to usual care; therefore, the benefits achieved were associated with cost savings. In another study, Romeo et al. [78] found that the WHELD intervention was cost-effective compared to usual care alone across a wide range of WTPs on the part of a decision maker for a unit improvement in outcome (the considered outcome measures were both QALYs and CMAI scores). The cost-effectiveness was mostly attributed to the lower health and social care costs faced by the intervention group compared to the control group. The authors also found that these results were mainly relevant to residents with clinically significant agitation in dementia. These studies on WHELD [77, 78] aimed to report on the value of nursing home residents using interventions that consider the reduction of antipsychotic use, but they did not find a significant reduction in antipsychotic consumption in the treated group of PwDs compared to the control group.

Steinbeisser et al. [83] provide evidence of high cost-effectiveness for MAKS, a non-pharmacological treatment with four components (motor stimulation, activities of daily living stimulation, cognitive stimulation, and social functioning) for individuals in day care centres with mild or moderate dementia. They found that the intervention had a high probability to be dominant with better outcomes (a higher ability to perform activities of daily living measured with ETAM scores, better cognitive abilities measured with MMSE scores) and lower costs than usual care.

Other two multicomponent interventions [79, 84] show moderate levels of cost-effectiveness. Wolfs et al. [79] studied an intervention adopted in the Netherlands that consisted of an integrated approach protocol involving the use of a diagnostic research centre for psycho-geriatrics. This centre was designed to enable health professionals to first map the needs of the community-dwelling patient and their caregiver, and then deliver a personalised treatment course consisting of different kinds of activities. They found that the intervention was cost-effective in terms of QALYs gained but not in terms of improvements in clinical measures such as cognitive impairment or behavioural and psychological symptoms; however, these results were subject to relevant statistical uncertainty. El Alili et al. [84] focused on the Namaste Care Family Program for nursing home patients and they found limited evidence of cost-effectiveness owing to high statistical uncertainty surrounding the results and to the fact that for two outcome measures (QUALID for patient QoL and GAIN for caregiving) no societal WTP thresholds have been defined yet.

Lastly, one structured psychosocial intervention analysed within the Danish Alzheimer's Intervention Study (DAISY) [80] and one multicomponent support intervention for couples [81] showed a very low probability to be cost-effective compared to usual care even though neither was found to be detrimental for either patients or caregivers.

## Discussion

### Main findings

This systematic review highlights the main evidence on the cost-effectiveness of SC interventions for PwDs and their caregivers. The analysed studies were quite heterogeneous in quality and included relevant costs and outcome measures. Nevertheless, the higher quality studies may provide useful findings on the value for money of specific interventions.

Eleven studies provided evidence of high cost-effectiveness for seven interventions: two multicomponent programmes (WHELD, targeted towards patients in nursing homes [77, 78], and MAKS in day care centres for community-dwelling people [83]); two indirect interventions (a group living service for PwDs [67] and a community-based Dementia Care Management programme [63, 116]); two interventions, START and REACH II, aimed at caregivers of community-dwelling PwDs [71, 73, 117, 118]; and one community-based cognitive stimulation and occupational programme for community-dwelling PwDs [119]. None of these studies showed a low level of methodological quality as regards the economic analysis: five [63, 71, 83, 116, 119] were assessed as being of high quality according to the CHEC criteria [88] while six other studies [67, 73, 77, 78, 117, 118] received a medium-quality appraisal. Undoubtedly, both the target populations and the methods adopted for measuring the cost-effectiveness of these SC interventions are quite heterogeneous, as we have shown in the previous section. However, all the interventions with evidence of high cost-effectiveness except one (the REACH II intervention for informal caregivers analysed by Nichols et al. [73]) were also found to be dominant (less costly and more effective than comparators) with a high probability according to sensitivity analysis, pointing out that they appear to be particularly promising in terms of economic sustainability.

Other sixteen SC interventions were found to be moderately cost-effective: six cognitive stimulation and rehabilitation programmes for community-dwelling PwDs [121, 123, 127], nursing home residents [128], or PwDs in different settings [122, 134]; two home-based individually tailored physical exercise programmes [126, 135]; three indirect interventions, including a Dementia Care Management programme [62] and two home care services [70, 120]; three interventions for family caregivers [69, 76, 131]; and two multicomponent interventions targeted towards community-dwelling PwDs [79] and nursing home patients [84].

Twelve SC interventions showed low or no cost-effectiveness: two cognitive stimulation programmes directed at community-dwelling patient-caregiver dyads [124] and institutionalised patients [129]; one aerobic exercise and resistance training programme [125]; five indirect interventions, including training sessions on agitation for nursing home staff (MARQUE) [65], memory clinics [66], telecare [68] and Dementia Care Mapping in a nursing home setting [130, 133]; two interventions primarily targeted towards family caregivers [72, 132]; and two community-based multicomponent interventions [80, 81].

Our analysis partially confirms some results of previous systematic reviews. For example, the systematic reviews by Nickel et al. [42], Knapp et al. [43], and Clarkson et al. [46] suggested that tailored occupational therapy for community-dwelling patients and caregivers [119, 127] and cognitive and long-term psychological interventions directly delivered to PwDs [122, 134] may be either highly or moderately cost-effective with regard to specific outcomes, while joint reminiscence groups for PwDs and carers 124] were found unlikely to be cost-effective. Previous systematic reviews, including the analysis by Jones et al. [45], provided mixed evidence with regard to interventions aimed directly at informal caregivers, even though in more recent reviews [42, 46], the START intervention [71, 117, 118] emerged as potentially cost-effective. Our analysis shows that some forms of psychosocial intervention for informal caregivers are highly cost-effective (e.g., the START and REACH II programmes [71, 73, 117, 118]) or moderately cost-effective (e.g., the support programmes analysed by Drummond et al. [69] and Gaugler et al. [131], and the telehealth intervention considered by Shaw et al. [76]), while other similar interventions have little or no cost-effectiveness (e.g., family meetings and befriending [72, 132]).

In contrast to other reviews, we found a high value for money of structured multicomponent interventions targeted towards patients in nursing homes [77, 78] and individuals in day care centers with mild or moderate dementia [83], which have the potential to draw benefits from the most cost-effective one-dimensional programmes. This was the case for the UK-based Improving Wellbeing and Health for People with Dementia (WHELD) programme for patients in nursing homes, which combines person-centred care, physical exercise, psychosocial activities, training for care staff, and the development of a system for triggering the appropriate review of antipsychotic medications for PwDs [77, 78]. On the contrary, other multicomponent interventions were assessed as moderately cost-effective [79, 84] or without any significant cost-effectiveness [80, 81]. We also found evidence of moderate cost-effectiveness of specific cognitive stimulation programmes for institutionalised PwDs (in particular, the learning therapy programme analysed by Sado et al. [128]) or for community-dwelling PwDs and their carers (specifically, the tailored cognitive rehabilitation programme investigated by Clare et al. within the GREAT trial [121]). In contrast to previous reviews [42, 46], we were not able to find clear evidence of high value for money for exercise programmes. For example, two individually tailored exercise interventions significantly improved patients' physical performance [126] or NPI scores [135] but they did not appear cost-effective when considering QALY gains; the DAPA, another exercise programme that was studied by Khan et al. [125], was dominated by usual practice in terms of cost-effectiveness. Similarly, several indirect interventions centred around organisational and environmental changes showed either moderate [62, 70, 120] or no [65, 66] cost-effectiveness.

Finally, two studies [68, 76] provided inconclusive evidence regarding the cost-effectiveness of telehealth and telecare interventions to support independence and improve QoL of both patients and informal caregivers, which could be particularly useful in situations such as the coronavirus disease pandemic we are experiencing. While the FamTechCare intervention to assist informal caregivers appeared to be cost-effective when compared to traditional telephone support intervention [76], the use of assistive technology and telecare in supporting

PwDs to continue to live safely within their own homes [68] did not prove to be cost-effective compared to more basic systems mainly due to the difficulty of adapting the indirect intervention to the needs of PwD and their carers.

## Methodological and operational challenges for the cost-effectiveness of supportive care interventions

From this review, we can identify a number of critical issues concerning both the methodology of economic evaluations and actual barriers to achieving better value for money of SC for dementia.

A first issue is the high methodological heterogeneity of the available studies in terms of quality, populations studied (regarding severity, comorbidity, and care settings), and the inclusion of relevant costs and outcome measures, which make it difficult to generalise their results. A frequent limitation of the analysed studies is the short time frame adopted for measuring most outcomes and costs. This is a result of the fact that most studies were trial-based evaluations. Additional research would be required to investigate the effects of SC over longer time horizons. For example, using a decision analysis modelling strategy to compare the costs and effectiveness of the interventions in the longer term could be an option to explore.

A second methodological issue concerns the instruments used to measure the outcomes of SC in terms of QoL for people with dementia and to derive QALYs in cost-utility analyses. The reviewed studies applied two types of instruments: a) generic instruments to assess HRQoL, such as the EQ-5D [62, 65, 66, 68, 79, 80, 84, 122, 124–126, 128, 133], the IWB scale [67, 70], and the SF-12 [63, 72, 116]; and b) dementia-specific instruments to measure the QoL of PwDs, such as the QoL-AD [117, 122–125, 134], the DEMQOL [121, 122], the DEMQOL-Proxy [65, 77, 78, 122, 123, 133, 135], and the QUALID [84]. The generic and dementia-specific QoL measures generally did not provide consistent cost-effectiveness findings. In particular, generic measures may not capture all relevant aspects associated with PwDs' experiences, even though instruments such as the SF-12 or EQ-5D have been shown to be suitable for HRQoL self-reporting by PwDs in mild and moderate stages [63, 66, 68, 116, 122, 124–126].

Since the assessment of self-report HRQoL and dementia-related QoL in PwDs is often characterised by recall bias and missing values, many studies opted for using carer-proxy reports of PwD QoL [62, 65, 68, 72, 77–80, 84, 117, 122–125, 133, 135]; another option was to convert data from clinical/health measures into QoL estimates according to the views of expert external observers [67, 70]. However, there are important differences between self-report and carer-proxy (or expert-proxy) reports [42, 137, 138]. For this reason, it may be useful to use both self and proxy ratings in the economic analyses of SC. Actually, some of the analysed studies [68, 122–125, 133] provided both measures of PwDs' QoL, thus confirming that self-rated and proxy QoL measures often have low levels of overall agreement and therefore cannot be assumed to substitute for each other. Furthermore, because the outcomes for caregivers and PwDs are typically interlinked, it is important to jointly assess the QoL of patient-caregiver dyads in order to take into account the type of caregiving relationship, which is an area of outcome assessment that has not yet been adequately developed. Four studies considered in our review assessed the QoL of patient-caregiver dyads [62, 66, 72, 80] by calculating the combined QALY scores through the simple summing of the QALYs for the PwD and the caregiver.

Another methodological issue concerns the identification and evaluation of the costs of SC interventions. In this respect, 18 out of 39 of the studies under review adopted a narrow perspective, looking only at the health and social care system and overlooking the opportunity costs of informal caregiver inputs and the impacts of caring on their own health and wellbeing.

Given the pivotal roles of family and other carers in dementia care, several authors [42, 43] recommend that economic evaluations of SC programmes for PwDs take on a societal perspective, including all relevant costs irrespective of where they occur and where they are funded.

Regarding the operational challenges surrounding the adoption of cost-effective SC, a first issue to consider is that pure cost-effectiveness analyses measure the ICER as an incremental cost per one-point difference in specific outcome measures, such as the MMSE [83, 122, 134], QoL-AD [117, 122, 124, 134], NPI [62, 135], CMAI [78, 129, 133], or carer's HADS scores [71, 117, 118]. However, in contrast to a cost-utility analysis, where the ICER is expressed in terms of the incremental cost per QALY gained and the British NICE's acceptability threshold range of £20,000–30,000 per QALY is frequently used, no established cost-effectiveness benchmark exists for such outcome changes. Therefore, as reported by many of the analysed studies, it is quite difficult to ascertain whether a particular SC intervention represents good value for money to the health and social care system, because we do not know the decision maker's willingness-to-pay for a one-unit reduction in the MMSE, Qol-AD, NPI, CMAI, or HADS scores. In these cases, a useful strategy adopted by some of the studies examined [62, 63, 83] could be to measure the probability that the intervention leads to a significant outcome improvement for different levels of the societal WTP, starting from a WTP equal to zero (i.e. without entailing increased costs for the taxpayer). Another often used strategy is to measure the societal WTP by quantifying the costs avoided thanks to the improvement in the outcome. For example, Murman et al. [139] showed that a 1-point worsening of the NPI score is associated with an incremental increase of between $247 and $409 per year in total direct costs of care based upon year 2001 US dollars. However, even in the absence of a WTP defined a priori by the policymaker for improvements in some of specific outcomes (MMSE, QoL-AD, NPI, etc.), carrying out comparative analyses of cost-effectiveness such as ours can provide useful policy indications, highlighting, for homogeneous categories of interventions and PwDs, those types of SC which show lower ICERs. For example, in previous section, we have shown that, after controlling for characteristics of targeted population (patients with mild-to-moderate dementia in different settings), intervention duration and price inflation, the cognitive stimulation therapy studied by Knapp et al. [134] had much lower ICERs than the maintenance cognitive stimulation therapy analysed by D'Amico et al. [122] in terms of both MMSE points and Qol-Ad points gained.

A final operational challenge underlined also by Knapp et al. [43] is that the cost-effectiveness of SC for PwDs depends crucially on the degree of integration between health and social care services, which are often delivered by different providers and funded from different budgets. It is therefore crucial to improve coordination between these services in order to increase the efficiency and effectiveness of interventions for PwDs.

## Limitations

Even though we conducted a comprehensive literature search based on extensive search terms, some papers meeting the criteria for inclusion might not have been identified. Furthermore, our systematic review may be subject to a language bias, as only publications in English or those with an abstract in English were included.

Some studies reviewed [67, 69, 70] were published years before the development of guidelines for assessing the methodological quality of health economics evaluations, such as the CHEC criteria [88]; therefore, their quality assessment may have been compromised. In any event, we decided to include these studies in the review because they evaluated forms of home support not considered by more recent studies, thereby allowing us to cover a wider range of SC interventions for PwDs.

## Future research

Our systematic review has highlighted the potential cost-effectiveness of multicomponent SC interventions targeted towards patients in nursing homes (e.g., the WHELD programme [77, 78]) that combine several interventions (person-centred care, physical exercise, psychosocial activities, behaviour management and training for care staff, the development of multi-disciplinary teams) with positive effects in terms of QoL and a decrease in challenging behaviour. Other studies [79–81] provided mixed evidence with regard to the value for money of community-based structured multicomponent interventions targeted at persons with MMD. Future research should therefore examine the cost-effectiveness of structured multicomponent interventions in different care settings and consider subgroups of PwDs at different disease stages. Furthermore, it is important to assess the impact on cost-effectiveness of the different components of multi-disciplinary interventions by focusing on the assessment of the roles of care coordination and case management. Eliciting the contribution of each component to the interventions' costs and outcomes would be important in terms of policy by highlighting how and why specific interventions may work to benefit PwDs and/or their caregivers.

Similar to other systematic reviews [42, 45, 46], we found mixed evidence with regard to SC interventions targeted towards informal caregivers. Specifically, some forms of psychosocial intervention for informal caregivers are highly cost-effective [71, 73, 117, 118] or moderately cost-effective [69, 76, 131], while others have shown little or no cost-effectiveness [72, 132]. As such, further investigation is needed to ascertain the real effects of interventions aimed directly at carers.

Finally, given that, as mentioned in the Introduction, one of the key aspects of SC should be the decreasing reliance on medications, in particular antipsychotics, that do not offer a sufficient benefit relative to the risks they pose, it appears somewhat surprising that only a few of the examined studies considered the reduction in antipsychotic drug consumption as one of the outcomes to be evaluated. In our opinion, future cost-effectiveness analyses of SC interventions should focus more on this aspect which so far appears to be rather neglected.

## Conclusion

To assess the current state of research on the cost-effectiveness of SC interventions for dementia, we performed a systematic review of the economic evidence, which is still scarce despite the several calls for action that have been made in the past few years [19, 43, 45, 140]. We reviewed 39 studies that analysed 35 SC programmes located at different stages of the care pathway for dementia that were generally directed at patient-caregiver dyads. Most interventions (23 out of 29) were implemented in European countries with comparable underlying health and social care systems.

We found that the most promising SC programmes in terms of cost-effectiveness were some multicomponent interventions targeted towards both nursing home residents, such as the WHELD programme [77, 78] and day-care service users [83], together with some forms of tailored occupational therapy [119] and home care support services [63, 67, 116] for community-dwelling PwDs. Our analysis has also shown that some forms of psychosocial intervention for informal caregivers of community-dwelling PwDs, such as the REACH II and START programmes [71, 73, 117, 118], were highly cost-effective. These results suggest the importance of policies promoting the adoption of effective supportive care interventions to increase the economic sustainability of dementia care.

Further research is required to establish the cost-effectiveness of structured multicomponent interventions in different care settings by considering subgroups of PwDs at different disease stages and assessing the impact of each component of the intervention. Moreover, since

the evidence on the cost-effectiveness of SC interventions targeted towards informal caregivers is mixed, further investigation is needed to ascertain the real effects of these interventions on both the PwD and his/her carer. Lastly, we think that empirical evidence on the real ability of SC interventions to reduce the use of antipsychotic medications in PwDs is still lacking and should be considered in future research.

## Supporting information

**S1 Checklist. PRISMA 2020 checklist.**
(DOCX)

**S2 Checklist. PRISMA 2020 for abstract checklist.**
(DOCX)

**S1 File. Electronic search strategy.**
(DOCX)

**S1 Table. Assessment of methodological quality of the studies on cognitive therapy interventions.**
(DOCX)

**S2 Table. Assessment of methodological quality of the studies on physical activity interventions.**
(DOCX)

**S3 Table. Assessment of methodological quality of the studies on indirect strategies.**
(DOCX)

**S4 Table. Assessment of methodological quality of the studies on interventions primarily aimed at supporting family caregivers.**
(DOCX)

**S5 Table. Assessment of methodological quality of the studies on multicomponent interventions.**
(DOCX)

## Acknowledgments

We thank Francesco Miele and Federico Neresini for their useful comments.

## Author Contributions

**Conceptualization:** Angelica Guzzon, Vincenzo Rebba, Giovanni Boniolo.

**Data curation:** Angelica Guzzon, Vincenzo Rebba.

**Formal analysis:** Angelica Guzzon.

**Investigation:** Angelica Guzzon, Vincenzo Rebba.

**Methodology:** Angelica Guzzon, Vincenzo Rebba, Omar Paccagnella, Michela Rigon, Giovanni Boniolo.

**Project administration:** Vincenzo Rebba.

**Resources:** Vincenzo Rebba, Giovanni Boniolo.

**Supervision:** Vincenzo Rebba, Giovanni Boniolo.

**Validation:** Vincenzo Rebba, Omar Paccagnella, Michela Rigon, Giovanni Boniolo.

**Visualization:** Angelica Guzzon, Vincenzo Rebba.

**Writing – original draft:** Angelica Guzzon, Vincenzo Rebba, Omar Paccagnella.

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
