## [Decision Letter · Decision Letter 0]

6 Sep 2021

PONE-D-21-04004Less drugs and more care: A systematic review of cost-effectiveness of supportive care interventions for dementiaPLOS ONE

Dear Dr. Rebba,

Thank you for submitting your manuscript to PLOS ONE. After careful consideration, we feel that it has merit but does not fully meet PLOS ONE’s publication criteria as it currently stands. Therefore, we invite you to submit a revised version of the manuscript that addresses the points raised during the review process.

This paper focused on an important subject. It is a practical work and can be used by manager and policy maker to take good decision regard dementia care services. The reviewers have raised a number of points which we believe would improve the manuscript. In addition to the items raised by the reviewer, please address the following points before more consideration:

This manuscript could benefit from a reorganization and a few more summary tables.  As it is written, it does not provide a clear and useful message.There is an updated PRISMA 2020 which should be used rather than the 2009 version. Please submit your revised manuscript by Oct 21 2021 11:59PM. If you will need more time than this to complete your revisions, please reply to this message or contact the journal office at plosone@plos.org. Please include the following items when submitting your revised manuscript:A rebuttal letter that responds to each point raised by the academic editor and reviewer(s). You should upload this letter as a separate file labeled 'Response to Reviewers'.A marked-up copy of your manuscript that highlights changes made to the original version. You should upload this as a separate file labeled 'Revised Manuscript with Track Changes'.An unmarked version of your revised paper without tracked changes. You should upload this as a separate file labeled 'Manuscript'.

We look forward to receiving your revised manuscript.

Kind regards,

Kamal Gholipour, PhD

Academic Editor

PLOS ONE

Journal Requirements:

3. PLOS requires an ORCID iD for the corresponding author in Editorial Manager on papers submitted after December 6th, 2016. Please ensure that you have an ORCID iD and that it is validated in Editorial Manager. To do this, go to ‘Update my Information’ (in the upper left-hand corner of the main menu), and click on the Fetch/Validate link next to the ORCID field. This will take you to the ORCID site and allow you to create a new iD or authenticate a pre-existing iD in Editorial Manager. Please see the following video for instructions on linking an ORCID iD to your Editorial Manager account: https://www.youtube.com/watch?v=_xcclfuvtxQ.

Reviewers' comments:

Reviewer's Responses to Questions

**Comments to the Author**

1. Is the manuscript technically sound, and do the data support the conclusions?

Reviewer #1: No

Reviewer #2: Partly

2. Has the statistical analysis been performed appropriately and rigorously? 

Reviewer #1: No

Reviewer #2: N/A

3. Have the authors made all data underlying the findings in their manuscript fully available?

Reviewer #1: Yes

Reviewer #2: Yes

4. Is the manuscript presented in an intelligible fashion and written in standard English?

Reviewer #1: Yes

Reviewer #2: Yes

5. Review Comments to the Author

Reviewer #1: Thank you for inviting me to review this manuscript. This paper reviews the global evidence on cost-effectiveness of supportive care interventions for dementia. This study engages massive works as the authors include five categories of supportive care intervention for both patients and their caregivers within total of 3,22 papers identified and 33 papers included in analysis. Also, this review encompasses too many outcomes (nine for patients living with dementia and eleven outcomes for caregivers). High volume of outcomes prevents deeper analysis and insights (i.e., meta-analysis of cost effectiveness). Here are my comments in detail:

1. Abstract: What does it mean SCIs in the abstract? Please provide full terminology first.

2. Abstract: Please clarify how many databases used? “Major” is vague.

3. Abstract: Please clarify publication period.

4. Abstract: Protocol registration missing

5. Introduction: Abbreviation of SC means supportive care (Line 83), then, unclear abbreviation of SCIs (Line 107, 177). Please use consistent term.

6. Introduction - Line 168: Please provide reference source.

7. Introduction - Line 184 to 250 should be briefly present in a table to help readers summarize main intervention. This table should be placed in the method under (I-Intervention question).

8. Method: Lack of PICO(S) questions. What type of interventions were mentioned (RCT, CCT, time series, Clinical study reports?). I suggested that the authors focus on the outcome related to cost-effectiveness and economic evaluation to match with the study objective.

9. Method - Line 256: Please clarify publication period and why this period was selected.

10. Method - Search term might not reflect the whole global literature on dementia SCIs. Please re-searching with more specific terms such as: “dementia” or “cognitive impairment” or “Alzheimer” or “senile". Intervention should be included as well [e.g “intervention” or “therapy” or “treatment” or “support” or or “education” or “psychoeducation” or “cognitive behaviour therapy or “psychotherapy”]

11. Method - Database: only two databases MEDLINE (PubMed) and CDSR (Cochrane Database of Systematic Reviews. It is recommended to search widely on Cochrane Central Register of Controlled Trials (CENTRAL) and MEDLINE, together with Embase, PsycINFO.

12. Method- The paper has not shown any effort to access grey literature.

13. Method: Please provide information on exclusion criteria (e.g. was the conference abstract/ thesis included).

14. Method – data extraction: Unclear how the data were extracted and assessed.

15. Inconsistent figures: Abstract mentions 3,221 articles retrieved while line 369 states that 3,218 publications identified.

16. Method – Outcome: Many outcomes mentioned but the outcome related to cost – effectiveness is unclear (usual care/.

17. Method – Comparison: Unclear the characteristics of control groups.

18. This manuscript shows minimum effort for assess publication bias as three papers were included after screening the references of the primary studies. I am wondering how good the search team can be performed and how many studies were not included due to insufficient search strategy.

19. Line 381: The authors reviewed 33 studies that analysed 29 interventions, what type of study design of 4 remaining studies? This statement might not fit with the total 31 trials, line 405-406, which described 26 RCT plus 5 non-randomised comparisons (CCT may be). Please avoid inconsistent figures.

20. Finding section and Table 3, 4, 5: The current structure is very confused and repetition when reporting by level of cost-effectiveness (high, moderate, and low). It doesn’t make sense to compare from different intervention (e.g. two cognitive trials in high cost effectiveness group vs. six moderate cost-effective group vs. one low effective group). Also, the authors have mixed up the trials that targeted for caregivers and PwD. This section and three tables need to consider primarily synthesizing findings based on type of intervention and target population. The readers would not interest in long description and intervention repetition.

21. Minimum effort to take into account of risk of bias in assessing effectiveness (e.g GRADE).

22. Discussion – Mai findings: the authors highlight the number of studies with high cost-effectiveness, but they are non-comparable. How can you conclude that SCIs focusing on cognitive intervention is cost-effective if two studies with high cost-effectives vs. six moderate studies)?

23. Based on evidence provided, I have not convinced by your title: “less drugs and more care for PwDs”. The reason come from most control groups in the primary trials in your review were usual care/ wait list/ non-pharmacotherapy. The title should be revised in caution to fit with evidence provided.

Reviewer #2: This work is a systematic review of cost-effectiveness studies for the non-pharmacologic treatment of dementia. The authors have conducted a large amount of work to identify 33 publications. The review and description of the studies is solid. What is lacking for me as a reader is the overall significance of the findings and general insights from the review. That is the work is a compilation and description of the reporting and conduct of the CEMs, where as a synthesis and interpretation of the results would be more interesting and informative.

Some general comments:

* The introduction is very long. The authors make the point that pharmacotherapy is not a sufficient treatment for the economic and humanistic burden of dementia, however an entire page is devoted to describing drug therapies. This was particularly confusing when the SLR methods specifically say only non-pharma studies were included. I think the introduction should be more concise and present the specific arguments to motivate research question of this work.

* The research question was not totally clear other than to compile this collection of studies. Based on what was presented there are a few directions this study can go. 1) What is the CE and the drivers of value of the non-pharma therapies for dementia? This would include an assessment of the cost components, perspectives, populations, outcomes etc., synthesized in a way that from the collective evidence a few key themes emerged. 2) What are the various methods for calculating the value non-pharma therapies? This would be a similar synthesis of the methods, and how different methods may or may not influence the result. 3) How is CE reported among the literature? There are a lot of various ICERs calculated and some discussion of the differences between them the pros/cons/situations in which they are appropriate could be interesting.

* The conclusion even mentioned that "evidence was mixed" but there are a lot of reasons for differing conclusions of CE analyses. This should be investigated and reported.

Overall, I do hate it myself when reviewers suggest a completely different analysis than what I submitted, but in the case of this article the structure of publication-description-publication-description is the least informative and least interesting way to describe this body of data. The introduction seemed that the motivation for this work is to show that non-pharma therapies have more holistic value than pharma-therapies. However, the data presented in the way that presented did not communicate that at all.

Specific comments

* The pubs were divided into high, moderate, and low value. Why? What was the threshold used to make this decision? Was there anything in the analysis or perspectives that caused them to be in these buckets?

* The methods contain a lot of basic health economic explanation. This can be removed.

* Some edits provided in PDF emailed

* The authors mentioned data were to heterogeneous to support a meta-analysis. Why would one want to meta-analyze results of CE studies?

6. PLOS authors have the option to publish the peer review history of their article (what does this mean?). If published, this will include your full peer review and any attached files.

Reviewer #1: No

Reviewer #2: No

---

## [Author Response · Author response to Decision Letter 0]

28 Feb 2022

RESPONSE TO EDITOR

Thank you very much for your reply and the invitation to revise the manuscript for PLoS ONE. 

As requested, we have updated the manuscript to reflect the change from PRISMA 2009 to PRISMA 2020. Moreover, we have reorganised the manuscript adding a few more summary tables.

With regard to the “data availability statement”, we would like to clarify that the data repository information we could provide at acceptance concerns the pdf versions of the studies that have been analysed in the systematic review.

We enclose the new version of the manuscript with the revised title “The value of supportive care: A systematic review of cost-effectiveness of non-pharmacological interventions for dementia” (to meet a Reviewer’s request). Moreover, we extended our search strategy to meet the helpful suggestions provided by a Reviewer and we also took the chance to update the review to include suitable studies published between our first and second submission (now we perform the systematic review between February 2019 and December 2021). This allowed us to include in the analysis six additional relevant studies.

We think that the new version addresses the points raised during the review process.

RESPONSE TO REVIEWER 1

We thank the Reviewer for providing valuable feedback that really helped to improve our paper.

We enclose the new version of the manuscript with the revised title “The value of supportive care: A systematic review of cost-effectiveness of non-pharmacological interventions for dementia” (to meet the Reviewer’s request). Moreover, we extended our search strategy to meet the helpful suggestions provided by the Reviewer and we also took the chance to update the review to include suitable studies published between our first and second submission (now we perform the systematic review between February 2019 and December 2021). This allowed us to include in the analysis six additional relevant studies.

We think that the new version addresses the points raised during the review process. Below is a point-by-point response to the Reviewer’s comments.

1. [Abstract: What does it mean SCIs in the abstract? Please provide full terminology first]

We clarified the acronym in the abstract.

2. [Abstract: Please clarify how many databases used? “Major” is vague]

Trying to be mindful of the word limit given for abstracts, we clarified in the abstract the databases used.

3. [Abstract: Please clarify publication period]

We added information regarding the publication period considered in the abstract.

4. [Abstract: Protocol registration missing]

No prespecified protocol was followed for this systematic review. We clarified this aspect in the manuscript in sub-section “Search strategy and criteria for inclusion” (on page 9).

5. [Introduction: Abbreviation of SC means supportive care (Line 83), then, unclear abbreviation of SCIs (Line 107, 177). Please use consistent term]

We made sure to use the aforementioned abbreviations consistently throughout the article.

6. [Introduction - Line 168: Please provide reference source]

We added references to back up that particular claim. On page 4 of the revised manuscript we report the references [24-27] on the link between the use of antipsychotic drugs in dementia patients and an increase in the risk of acute pulmonary diseases, hip fracture, thromboembolism, and stroke.

7. [Introduction - Line 184 to 250 should be briefly present in a table to help readers summarize main intervention. This table should be placed in the method under (I-Intervention question)]

We have summarised most of the information originally included between lines 184-250 in table 1 in the section “Materials and Methods” (on pages 7-8).

8. [Method: Lack of PICO(S) questions. What type of interventions were mentioned (RCT, CCT, time series, Clinical study reports?). I suggested that the authors focus on the outcome related to cost-effectiveness and economic evaluation to match with the study objective]

We were not deliberately following PICO principles while working on our manuscript because we believe they would have been redundant given that PRISMA recommendations for systematic reviews ask for the same type of information. However, we have provided information with respect to population, type of intervention, control groups, outcomes, etc. for all the studies considered in Tables 4, 5, 6, 7 and 8 in the section “Results”. 

9. [Method - Line 256: Please clarify publication period and why this period was selected]

The publication period originally considered was from inception to March 2020, but we recognise we could have been more explicit when describing the timeframe considered and now (in the abstract and in sub-the section titled “Search strategy and criteria for inclusion”, on page 9) we provide additional information. We also took the chance to update the review to include suitable studies published between our first and second submission. Therefore, now we perform the systematic review between February 2019 and December 2021 and clarify that we consider studies published through December 2021 with no lower date limit.

10. [Method - Search term might not reflect the whole global literature on dementia SCIs. Please re-searching with more specific terms such as: “dementia” or “cognitive impairment” or “Alzheimer” or “senile". Intervention should be included as well [e.g “intervention” or “therapy” or “treatment” or “support” or or “education” or “psychoeducation” or “cognitive behaviour therapy or “psychotherapy”]

We clarified our search strategy in the sub-section “Search strategy and criteria for inclusion” (on page 9). As detailed also in the S1 File in the Appendix, we extended our search strategy to include the helpful suggestions provided and kept the search field intentionally wide to minimise the risk of not capturing relevant articles. This allowed us to find some additional relevant studies. In particular, we included in the analysis six recent articles, some of which were published after the first submission. Moreover, we do note that the new search strategy was able to capture the three articles that in the previous version of our manuscript were reported as having been included after screening the references of other articles.

11. [Method - Database: only two databases MEDLINE (PubMed) and CDSR (Cochrane Database of Systematic Reviews. It is recommended to search widely on Cochrane Central Register of Controlled Trials (CENTRAL) and MEDLINE, together with Embase, PsycINFO]

We extended our search strategy to include the databases suggested here, but no additional relevant results were found compared to our initial search. With the exception of the six most recent articles identified, the inclusion of the databases we had not included in our initial search strategy mainly led to duplicates, as reflected in the numbers reported in the flowchart included in the article on page 15 (5,479 versus 3,218 in the first version of the manuscript). The fact that the inclusion of the additional databases only led to duplicates is not surprising, as most of them rely on MEDLINE data, e.g. Embase (which leases MEDLINE data through a license). A source for this claim can be found at this link: https://www.nlm.nih.gov/databases/journal.html.

12. [Method- The paper has not shown any effort to access grey literature]

The databases considered cover – at least in part – grey literature in addition to regular literature, and indeed one of the studies considered (Van de Ven et al., 2014 [130]) originated as part of a PhD thesis while other three studies (Clare et al., 2019) [121]; Orgeta et al., 2015 [123], Woods et al., 2012 [124]) were technical NIHR HTA programme reports (www.hta.ac.uk).

13. Method: Please provide information on exclusion criteria (e.g. was the conference abstract/ thesis included]

We have provided information on inclusion and exclusion criteria in the sub-section “Search strategy and criteria for inclusion” (on page 9).

14. [Method – data extraction: Unclear how the data were extracted and assessed]

We added to the manuscript more information on the process of data extraction and assessment (in the sub-section “Data collection and analysis” on page 10).

15. [Inconsistent figures: Abstract mentions 3,221 articles retrieved while line 369 states that 3,218 publications identified]

We made sure to avoid inconsistencies in the revised manuscript.

16. [Method – Outcome: Many outcomes mentioned but the outcome related to cost – effectiveness is unclear (usual care/]

Now in the section “Results” (sub-section Evidence of cost-effectiveness of supportive care interventions from reviewed studies”) we provide more clear information on outcomes and on the assessment of cost-effectiveness of the analysed interventions, following the criteria detailed on page 18. In tables from 4 to 8 we specify the comparators and the outcomes used to develop the cost-effectiveness analysis in the considered studies. 

17. [Method – Comparison: Unclear the characteristics of control groups]

Information on the control groups considered in each study have been added to the descriptive tables 4, 5, 6, 7 and 8 in the section “Results” (sub-section Evidence of cost-effectiveness of supportive care interventions from reviewed studies”). In the vast majority of the articles considered the intervention is compared to “care as usual”, therefore it is reasonable to assume – and indeed all the articles make sure this is the case – the intervention and control groups have statistically non-significant differences in characteristics ex ante.

18. [This manuscript shows minimum effort for assess publication bias as three papers were included after screening the references of the primary studies. I am wondering how good the search team can be performed and how many studies were not included due to insufficient search strategy]

As previously detailed in the answers to points 10 to 12, we have no reason to believe our search strategy is insufficient. We note in the “Limitations” sub-section (on page 54) the possibility of a language bias due to the fact that only publications in English or those with an abstract in English were included.

19. [Line 381: The authors reviewed 33 studies that analysed 29 interventions, what type of study design of 4 remaining studies? This statement might not fit with the total 31 trials, line 405-406, which described 26 RCT plus 5 non-randomised comparisons (CCT may be). Please avoid inconsistent figures]

In the revised manuscript we have considered 39 studies, and some of these consider the same intervention but in a different setting or with a different timeframe. This is the reason why there are more studies (39) than interventions (35). We have added clarifications with respect to these aspects in the section “Results” (sub-section “Characteristics of the included studies” on pages 15-16).

20. [Finding section and Table 3, 4, 5: The current structure is very confused and repetition when reporting by level of cost-effectiveness (high, moderate, and low). It doesn’t make sense to compare from different intervention (e.g. two cognitive trials in high cost effectiveness group vs. six moderate cost-effective group vs. one low effective group). Also, the authors have mixed up the trials that targeted for caregivers and PwD. This section and three tables need to consider primarily synthesizing findings based on type of intervention and target population. The readers would not interest in long description and intervention repetition]

We do agree this section can benefit from more conciseness and has been shortened. We also took the advice to divide interventions according to their type rather than their level of cost-effectiveness, hoping to draw more informative conclusions from this effort.

On the other hand, we do not believe we are “mixing up” trials targeting PwDs versus caregivers, as we are classifying interventions explicitly aimed at caregivers into their own category. We also note that in this field of research it is not always possible to have a clear-cut separation between interventions targeting PwDs and interventions targeting caregivers, as oftentimes studies focus on the patient-caregiver dyad.

21. [Minimum effort to take into account of risk of bias in assessing effectiveness (e.g GRADE)]

We have provided information on how the quality of the studies considered was assessed in the sub-sections “Quality appraisal of included studies” (on page 10) and “Quality assessment of the included studies” (on page 17). An in-depth analysis can also be found in the Tables S1-S5 in the Appendix to the article. The quality assessment table we are using (CHEC) is more extensive and better suited for economic studies compared to GRADE, and gauges not only the quality of the evidence, but also the quality of a study as a whole.

22. [Discussion – Main findings: the authors highlight the number of studies with high cost-effectiveness, but they are non-comparable. How can you conclude that SCIs focusing on cognitive intervention is cost-effective if two studies with high cost-effectives vs. six moderate studies)?]

Given the reorganisation of the article as described in the answer to point 20 and the definition of the criteria to assess cost-effectiveness of different studies reported on page 18 (sub-section “Evidence of cost-effectiveness of supportive care interventions from reviewed studies”), we think that now the summary of the main findings in the “Discussion” and “Conclusion” sections should be clearer.

23. [Based on evidence provided, I have not convinced by your title: “less drugs and more care for PwDs”. The reason come from most control groups in the primary trials in your review were usual care/ wait list/ non-pharmacotherapy. The title should be revised in caution to fit with evidence provided]

We have accepted this valuable suggestion and now we have changed the title of the manuscript.

RESPONSE TO REVIEWER 2

We thank the Reviewer for providing valuable feedback that really helped to improve our paper.

We enclose the new version of the manuscript with the revised title “The value of supportive care: A systematic review of cost-effectiveness of non-pharmacological interventions for dementia”. Moreover, we extended our search strategy to meet the helpful suggestions provided by a Reviewer and we also took the chance to update the review to include suitable studies published between our first and second submission (now we perform the systematic review between February 2019 and December 2021). This allowed us to include in the analysis six additional relevant studies.

We think that the new version addresses the points raised during the review process. Below is a point-by-point response to the Reviewer’s comments.

A. [The introduction is very long. The authors make the point that pharmacotherapy is not a sufficient treatment for the economic and humanistic burden of dementia, however an entire page is devoted to describing drug therapies. This was particularly confusing when the SLR methods specifically say only non-pharma studies were included. I think the introduction should be more concise and present the specific arguments to motivate research question of this work]

The Introduction has been shortened, especially the section devoted to providing an overview of the state of the art with respect to pharmacological interventions.

B. [The research question was not totally clear other than to compile this collection of studies. Based on what was presented there are a few directions this study can go. 1) What is the CE and the drivers of value of the non-pharma therapies for dementia? This would include an assessment of the cost components, perspectives, populations, outcomes etc., synthesized in a way that from the collective evidence a few key themes emerged. 2) What are the various methods for calculating the value non-pharma therapies? This would be a similar synthesis of the methods, and how different methods may or may not influence the result. 3) How is CE reported among the literature? There are a lot of various ICERs calculated and some discussion of the differences between them the pros/cons/situations in which they are appropriate could be interesting]

Now, we present more clearly our research question in the Introduction, on pages 5-6.

We have completely reorganised the paper in the section “Results”, providing more informative and (we hope) useful details on the drivers of cost-effectiveness of the SC interventions analysed by the reviewed studies in tables 4, 5, 6, 7 and 8. These tables, and the comments provided in the new sub-section “Evidence of cost-effectiveness of supportive care interventions from reviewed studies” provide more specific information on targeted populations, outcomes and on the types of cost-effectiveness analysis performed in different studies.

We also clarify better the cost components, the perspectives, and the different methods for calculating the value non-pharma therapies in the reviewed studies in sub-section “Characteristics of the included studies” on pages 15-17.

Both in the new sub-section “Evidence of cost-effectiveness of supportive care interventions from reviewed studies” and in the revised sub-section “Methodological and operational challenges for the cost-effectiveness of supportive care interventions” we discuss extensively the differences between different ICERs and in particular the issues related the measurement of cost-effectiveness when the ICER is expressed in terms of specific outcome measures (e.g. MMSE, NPI or Qol-AD scores). 

C. [The conclusion even mentioned that "evidence was mixed" but there are a lot of reasons for differing conclusions of CE analyses. This should be investigated and reported. Overall, I do hate it myself when reviewers suggest a completely different analysis than what I submitted, but in the case of this article the structure of publication-description-publication-description is the least informative and least interesting way to describe this body of data. The introduction seemed that the motivation for this work is to show that non-pharma therapies have more holistic value than pharma-therapies. However, the data presented in the way that presented did not communicate that at all]

We have completely reorganised the manuscript to compare the cost-effectiveness of the interventions according to the type of intervention (cognitive, physical activity, indirect, aimed at caregivers, multicomponent). We think that the new sub-section “Evidence of cost-effectiveness of supportive care interventions from reviewed studies” (page 18) provides more useful information on the comparative value for money of different SC interventions.

D. [The pubs were divided into high, moderate, and low value. Why? What was the threshold used to make this decision? Was there anything in the analysis or perspectives that caused them to be in these buckets?]

Now we specify the criteria used to classify the cost-effectiveness of the interventions analysed by the reviewed studies at the beginning of the sub-section “Evidence of cost-effectiveness of supportive care interventions from reviewed studies” on page 18. 

E. [The methods contain a lot of basic health economic explanation. This can be removed]

That section has been completely reorganised and basic health economic explanations have been removed. 

F. [The authors mentioned data were too heterogeneous to support a meta-analysis. Why would one want to meta-analyze results of CE studies?]

Although possibly redundant, we believe it was necessary to specify why a meta-analysis was not possible as per PRISMA 2009 and PRISMA 2020 guidelines, which call for a meta-analysis whenever possible.

---

## [Editor Report · Decision Letter 1]

5 Feb 2023

PONE-D-21-04004R1

The value of supportive care: A systematic review of cost-effectiveness of non-pharmacological interventions for dementia

PLOS ONE

Dear Dr. Rebba,

Thank you for submitting your manuscript to PLOS ONE. After careful consideration, we feel that it has merit but does not fully meet PLOS ONE’s publication criteria as it currently stands. Therefore, we invite you to submit a revised version of the manuscript that addresses the points raised during the review process.

The authors did a nice job addressing comments and suggestions. However, I have a few minor additional comments.

Please correct and complete the title of tables as informative format that considered the full description of tables content.

We look forward to receiving your revised manuscript.

Kind regards,

Kamal Gholipour, PhD

Academic Editor

PLOS ONE
---

## [Author Response · Author response to Decision Letter 1]

11 Apr 2023

We would like to thank the reviewers again for spending their time evaluating our manuscript, and for providing valuable feedback that helped to further improve our systematic review. We have been able to incorporate changes to reflect the suggestions provided by the Editor and the reviewers.

As requested, we have corrected and completed the titles of tables 4, 5, 6, 7 and 8 in order to provide a full description of the tables content.

We think that the new version addresses the points raised during the review process.

Reviewer 1: We have incorporated all of your suggestiions into the revision. They were very helpful. Thank you. In particular, we have corrected and completed the titles of Table 4 (on page 21), Table 5 (on page 26), Table 6 (on page 31), Table 7 (on page 38), and Table 8 (on page 44). We made these changes in order to provide a full description of the tables content.

Reviewer 2: We have incorporated all of your suggestiions into the revision. They were very helpful. Thank you. In particular, we have corrected and completed the titles of Table 4 (on page 21), Table 5 (on page 26), Table 6 (on page 31), Table 7 (on page 38), and Table 8 (on page 44). We made these changes in order to provide a full description of the tables content.

---

## [Editor Report · Decision Letter 2]

20 Apr 2023

The value of supportive care: A systematic review of cost-effectiveness of non-pharmacological interventions for dementia

PONE-D-21-04004R2

Dear Dr. Rebba,

We’re pleased to inform you that your manuscript has been judged scientifically suitable for publication and will be formally accepted for publication once it meets all outstanding technical requirements.

Kind regards,

Kamal Gholipour, PhD

Academic Editor

PLOS ONE
---

## [Editor Report · Acceptance letter]

4 May 2023

PONE-D-21-04004R2 

The value of supportive care: A systematic review of cost-effectiveness of non-pharmacological interventions for dementia 

Dear Dr. Rebba:

I'm pleased to inform you that your manuscript has been deemed suitable for publication in PLOS ONE. Congratulations! Your manuscript is now with our production department. 

Kind regards, 

on behalf of

Dr. Kamal Gholipour 

Academic Editor

PLOS ONE